# Egalitarian norms can deflate identity-bias link in real-life groups

**Sami Çoksan** [1,2*], **Ahmed Faruk Sağlamöz**[3,4]

**1** Network for Economic and Social Trends, Western University, London, Ontario, Canada, **2** Department of Psychology, Erzurum Technical University, Erzurum, Türkiye, **3** Department of Psychology, Durham University, Durham, United Kingdom, **4** Department of Psychology, Ankara University, Ankara, Türkiye

* scoksan@uwo.ca

## Abstract

Social identity theory posits that group membership influences individual behavior by fostering a sense of belonging and promoting normative conformity within groups. While much research has shown a link between ingroup identification and ingroup bias, the role of ingroup norms in moderating this association remains less explored. Specifically, how varying norms (egalitarianism vs. favoritism) affect bias in individuals with high ingroup identification requires further investigation. To address this gap, we examined whether ingroup norms alter the strength of the identification-bias relationship in two studies ($N = 322$). We investigated how non-WEIRD real-life group members' ingroup bias was driven by their identification levels and perceived ingroup norm in Study 1 with a correlational design, and we experimentally manipulated ingroup norms in a simulated group discussion in Study 2. Both studies demonstrated that under a favoritism norm, participants with high ingroup identification showed greater ingroup bias, whereas this bias was deflated under an egalitarianism norm. However, contrary to our hypothesis, we did not find evidence that participants with high ingroup identification showed lower ingroup bias under the egalitarianism norm. We discuss these findings and suggest that fostering egalitarian norms within groups may reduce ingroup bias and discrimination, offering insight for interventions aimed at promoting intergroup harmony.

## Introduction

Social identities, such as membership to a family, ethnicity, gender, and religion, serve various functions in our lives: They have not only the capacity to provide meaning and purpose to our lives but also serve as a source of enhanced well-being by granting privileged access to emotional and material resources [1]. Building on such identities, when individuals are asked to allocate some resources between groups, they do so based on the recipient's identity in question. As such, simply perceiving others as belonging to certain outgroups may affect whether ingroup members

**Data availability statement:** All data and materials for both studies are publicly available on the Open Science Framework (OSF) at https://osf.io/2fha7/." now. The project is publicly available.

**Funding:** The author(s) received no specific funding for this work.

**Competing interests:** The authors have declared that no competing interests exist.

choose to allocate resources evenly or unevenly [2,3]. In such resource allocation tasks, the perception of resource scarcity appears to promote the endorsement of zero-sum beliefs and, in turn, favoring the ingroup over outgroups [4].

Since Social Identity Theory (SIT) was proposed to explain the role and importance of group-based identities in constituting the self-concept and regulating the behavior of group members [5], it has been repeatedly utilized to explore and account for ingroup bias [4,6,7]. Although it is argued that SIT does not clearly hypothesize a positive link between ingroup identification and ingroup bias [8,9], the existence of this link is well-documented [10–12]. However, the strength of this relationship depends on various factors, and one of those moderating factors is the ingroup norm [13,14].

### Ingroup identification and ingroup norms

Social psychologists have long studied the normative influence of group membership on the behavior of group members. This generated a line of research suggesting that behavioral uniformity performed by group members is acquired through conformity with the perceived ingroup norms [15–19]. For instance, Jetten and colleagues [14] found that people with stronger ingroup identification tend to be more motivated to conform to the ingroup norm than people with lower ingroup identification when the ingroup norm at play was differentiation, which refers to the tendency of individuals to distinguish their own group from other groups by emphasizing social, behavioral, or perceptual differences between them. Differentiation is also conceptually similar to favoritism, which refers to the tendency of individuals to evaluate their own group more positively than other groups, prioritize it in resource allocation, or grant it special advantages. The difference between people with high and low identification, however, disappeared when the fairness norm, which refers to collective standards within a group regarding what its members consider right, appropriate, or justified based on criteria such as merit, proportionality, or need, was in place. Fairness norms are thus context-dependent and may lead to unequal outcomes that are nonetheless perceived as legitimate or justified by group members. Another study showed that ingroup identification predicted intentions to adhere to ingroup norms for only those strongly identified with their group [20]. These findings indicate that people with high ingroup identification are expected to be more mindful of ingroup norms and accordingly revise their behavior than people with low ingroup identification.

In a group with egalitarian norms, people with high ingroup identification are more likely to adopt egalitarian behaviors by conforming to the ingroup norm, which may be due to their motivation to fit into the group [21], or their desire to be accepted within the group [22], to acquire social rewards such as signaling socially acceptable behaviors and strengthening their ties with the group [23], to maintain their reputation [24] and their social status within the group [25]. This might mean that conforming to ingroup norms (i.e., adopting more egalitarian behaviors) will strengthen their identification with and status within the group.

People with low ingroup identification tend to comply with group norms less than people with high ingroup identification. The weaker bond these members have with

the group may not sufficiently motivate them to exhibit behavior in line with the group norm because, for these members, conforming to group norms may not be as important or rewarding as for highly identified ingroup members [23]. Therefore, this diminished motivation helps explain the lower tendency to conform among less identified ingroup members [26,27]. However, this does not mean that people with low ingroup identification entirely reject group norms or that they show no ingroup bias. In fact, empirical research suggests that individuals with lower ingroup identification may still exhibit some degree of ingroup bias, although this bias tends to be weaker compared to that shown by high identifiers [13].

Likewise, low ingroup identification does not imply a complete absence of identification with the group. Members with low ingroup identification still tend to act in accordance with ingroup norms, albeit relatively weakly. In other words, even though their commitment and sense of belonging are lower, these members still tend to look to ingroup norms to determine the correct behavior in the social environment because these contexts provide strong guidance to members on what they should do. Thus, even individuals with low ingroup identification still follow these strong social cues to determine appropriate behavior in the social context, indicating a tendency to conform to group norms, though to a lesser extent than people with high ingroup identification [28]. However, the extent to which they comply with group norms depends on various external and situational factors rather than a strong intrinsic motivation to align with the group. For instance, social pressure and fear of exclusion [29], practical benefits of following norms (e.g., reducing cognitive effort in decision-making) [30], or maintaining future identity options [31] may all contribute to norm adherence even among those with low ingroup identification. Additionally, norm adherence can stem from cultural and structural influences, as seen in collectivist cultures where group compliance is emphasized regardless of personal identification [32]. Therefore, while individuals with low ingroup identification may still conform to group norms, their tendency to display ingroup bias is generally weaker than that of individuals with high ingroup identification. This could be because their group identity plays a less central role in shaping their behavior, making them less likely to favor their ingroup unconditionally. Alternatively, it could be that people with low ingroup identification adopt a more flexible and situational approach to social behavior, adjusting their responses based on external rewards, social expectations, or broader societal norms rather than a deeply internalized group loyalty.

Furthermore, how these norms are formed and how their impact is translated into behavior is not very clear. For instance, Iacoviello and Spears [7] have found that individuals tend to perceive that they are expected to favor their ingroup in both real and minimal groups. They argue that the fairness norm does not derive from intragroup processes, and it is often promoted and dictated by supra-ordinate elements that go beyond intergroup contexts. Accordingly, they suggest that there is a conflict between the external expectation of fairness imposed by outside *moral referees* and the internal expectation of favoritism imposed by the ingroup.

On that basis, the current research aims to create an internal expectation of egalitarianism, which refers to a social rule that demands the equal distribution of resources, opportunities, and treatment among members of different groups, imposed by the ingroup. Following Jetten and colleagues [14], McGuire and their colleagues [17], and Çoksan and Cingöz-Ulu [13], to the best of our knowledge, the current research is one of the few studies in the social identity literature that employs both a correlational (in Study 1) and an experimental design (in Study 2) to explore the effect of ingroup norm on ingroup bias and the ingroup identification-ingroup bias association in particular. In a recent study, Çoksan and Cingöz-Ulu [13], experimentally examined how Kurdish and Turkish participants behave under the influence of group norms (egalitarianism or favoritism). The findings showed that individuals with strong identification showed higher ingroup bias in the favoritism norm condition, but this relationship disappeared in the egalitarian norm condition. The theoretical contribution of their study lies in testing the moderating role of group norms on ingroup bias within contexts where real-life groups are involved in explicit conflict. In contrast, our studies aim to advance the social identity literature in three important ways. First, we sampled real group members (football fans) who are part of competitive groups, yet not involved in direct or violent conflict. Second, we focused on the egalitarianism norm, which emphasizes equal distribution, rather than the more flexibly defined fairness norm. Third, in Study 2, we manipulated group norms using a simulated group

discussion in which participants responded to the perceived opinions of their ingroup members and contributed their own views. While this setting is structured and sequential, and does not involve back-and-forth exchange, it mimics norm formation dynamics through exposure to a coordinated sequence of normative statements and prompted responses.

## Sampling real-life group members in a competitive context

We sampled non-WEIRD real-life group members (football team fans) contrary to many studies in the social identity approach; thus, we aim to extend the findings on the association between ingroup identification and ingroup bias to real and competitive groups in a non-conflictual context. In this way, we aim to enhance the cultural representativeness of the literature by filling a sampling gap, thereby bolstering the generalizability of findings and offering insights that may inform practical applications.

Unlike previous studies that examined intergroup bias in conflictual settings [13,14], where resource allocation occurs directly between two competing groups (e.g., Kurds vs. Turks), our study did not position participants in a direct intergroup competitive framework. Instead, participants allocated resources to their own group while the remaining funds were distributed among all teams in the league. Given that their team is not in realistic competition with the majority of these teams, this allocation process lacks the hallmarks of intergroup conflict. While competition among football teams exists, it does not constitute a conflictual intergroup setting in the way that ethnic or political rivalries do.

Moreover, in Türkiye, football fandom is a strong and consistently salient form of social identity. Fan identity plays an important social role in shaping social relations and group loyalties not only during matches but also in daily life such as reading about the matches and buying licensed merchandise [33]. For many individuals, loyalty to soccer teams creates a lifelong sense of belonging, and this identity is particularly strong among fans of major teams [34]. Therefore, the identity of football fans in Türkiye provides a relevant and powerful context for the study of group norms and identification processes.

## Fairness vs. egalitarianism

The way we operationalize the ingroup norm manipulation differs from Jetten and colleagues [14], who examined norms under fairness and favoritism conditions. A key issue with fairness is that it does not inherently promote equality between groups. In fact, fairness may paradoxically reinforce ingroup favoritism when it is perceived as a means to rectify past injustices [35]. Research also suggests that fairness judgments may be anchored in proportionality rather than strict equality [36]. That is, individuals tend to assess fairness based on the relative contributions of parties involved rather than an egalitarian principle that mandates equal division, leading to outcomes where inequality is perceived as fair.

Another factor that can shape fairness perceptions is status concerns, as higher-status group members are more likely to see unequal outcomes as fair when they believe their group is entitled to greater rewards [37]. This highlights that fairness can be self-serving as it enables high-status groups to justify and maintain inequalities while presenting them as fair [38]. Even in the absence of clear status differences, people still apply fairness flexibly to justify unequal distributions. For instance, in ultimatum game scenarios, individuals tend to offer unequal distributions while believing their offers are fair, despite the fact that recipients consider these allocations as unfair [39]. This finding demonstrates how fairness norms can be misaligned between groups, further reinforcing the idea that fairness can be self-serving and does not inherently promote equality.

Given that fairness is an ambiguous concept subject to multiple interpretations, including compensatory justice, proportionality, and status maintenance, it is challenging to ensure that fairness norms consistently lead to non-favoring distributions. This conceptual ambiguity has led researchers to replace fairness with egalitarianism as a more consistent, neutral, unambiguous and non-retaliatory framework for resource allocation [13]. By emphasizing strict equality in outcomes, egalitarian norms rule out context-dependent interpretations of fairness, which can be endorsed to justify ingroup favoritism.

Thus, our study chooses the norm of egalitarianism over fairness to avoid the unintended biases that fairness-based distributions can easily introduce. While fairness is a broader concept that can include various criteria such as merit, need, or proportionality, egalitarianism refers specifically to equal treatment and distribution regardless of status or contribution. By focusing on an unequivocally equal allocation of resources, we ensure that fairness, when interpreted in a way that permits ingroup favoritism via status reinforcement or retaliatory motives, does not undermine the impartiality of our norm manipulation.

## Manipulating the ingroup norm with a simulated group discussion

Lastly and most importantly, instead of presenting vignettes and describing ingroup norms, the manipulation task was designed around seemingly live discussions instructed by a moderator in Study 2. In other words, we aimed to eliminate the perception of ingroup norms as externally imposed in Study 2 by making the participants believe that they have a live discussion with other ingroup members. In a group of four football fans, the participants were always recognized last to speak, in line with Asch's classical studies [40] on normative social influence. Additionally, to avoid a possible negative effect of norm violation [41,42], normative influence was expected to be greater when participants spoke last. This was implemented using Qualtrics' simulated dialogue boxes created with JavaScript (see https://osf.io/2fha7/). The first three fans were nothing but predetermined scripts, one after the other speaking up seemingly as a response to previous opinions, all giving voice to the same policy that they believe their football team represents: equal or ingroup-favoring distribution of resources.

Moreover, the utilization of group discussion as a source of ingroup norm resonates deeply with Moscovici's [43] suggestion that group members come to an agreement on ingroup norms through actively engaging in discussions, as the adoption of group norms is not simply a one-sided transfer of group norm information. The main reason for choosing group discussions to manipulate norms in our research is the strong evidence that norms are dynamic processes created and altered through active participation rather than being rules passively received by group members [44]. Moscovici's insights on group discussions [43] explain that group norms are shaped and maintained not only through external guidance but also through ingroup interactions and participation. Group discussions show that group members not only accept existing norms but also collectively adopt them by negotiating and redefining them together. Therefore, in contrast to unilateral manipulations, this simulated method may lead to a stronger, sustained, and internalized adoption of norms [45,46]. The active participation of group members allows norms to create shared meanings among individuals and these norms become part of the group identity [45,47]. That is, the process of norm formation through group discussions reveals that group members are not passive recipients but active participants who shape norms. Therefore, the manipulation of ingroup norms in Study 2 is expected to support our hypotheses, since we expect more direct normative influence through group discussion than through the introduction of descriptive norms.

## Current study

Quite a few studies have examined how group members' behavior will differ if the ingroup norm indicates the opposite of what ingroup identification indicates, namely, less ingroup bias. Drawing on Reicher's [15] seminal study and Hunter and colleagues' [16] suggestion to take account of the context-dependency of the association between identity and action, what the current study aims to achieve is to capture a crucial factor, ingroup norm, the content of which can help explain some of the contextual variability in the ingroup identification and ingroup bias association. We focused on whether members with high identification in the egalitarianism ingroup norm, which indicates treating all groups equally, would show less ingroup bias because of aligning themselves with the egalitarian ingroup norm, or they would still favor their ingroup because of their high identification.

Based on this literature, throughout the two studies in the current research, we hypothesized that ingroup norm would moderate the association between ingroup identification and ingroup bias (H1). Specifically, we expected that when the ingroup norm is favoritism-oriented, participants would show greater ingroup bias as their ingroup identification increases (H2$_a$), whereas when the ingroup norm emphasizes egalitarianism, the relationship between ingroup identification and ingroup bias would be negative (H2$_b$). All data and materials for both studies are publicly available on the Open Science Framework (OSF) at https://osf.io/2fha7/.

## Study 1

### Method

The current study was approved by the Human Subjects Ethics Committee of Middle East Technical University, Ankara, Türkiye, with the number 219-ODTÜ-2019. Online written informed consent was obtained from all participants before participating in the study.

**Participants.** We reached 204 lay people, defined as individuals without specialized knowledge of social identity research but are naturally embedded in real-life group memberships as football fans, using snowball sampling. Since 16 participants identified the real purpose of the research (What do you think is the purpose of the research? An example of a participant's answer: The research aims to examine the relationship between group commitment, perceived group norm, and ingroup bias), they were excluded from the dataset before the analyses. The remaining 188 (75 female, 113 male) participants' mean age is 27.67 ($SD = 7.78$, 4 missing). Most of the participants are high school graduates ($N = 117$, 62.2%) or have a bachelor's degree ($N = 49$, 26%), classify themselves as on the middle socioeconomic level ($N = 100$, 53.2%), and fans of either Galatasaray ($N = 65$, 34.6%), Fenerbahçe ($N = 61$, 32.4%), or Beşiktaş ($N = 48$, 25.5%) football team. All three of these teams were competing to become champions in the Turkish Super League and their fans were in competition with each other, but these teams and their fans did not have the same level of intense competition with other teams in the league.

**Measurements.** We used a tailored version of Leach and colleagues' ingroup identification scale [48] adapted to Turkish by Balaban [49] to measure ingroup identification. The internal reliability of the 7-point Likert-type scale (1 = strongly disagree; 7 = strongly agree), consisting of 7 items (e.g., It is pleasant to be [football team name] fan), showed high internal reliability (standardized Cronbach's α = .94). A higher score indicates stronger identification with the ingroup.

One item was used to measure the perceived ingroup norm. The item directly asks whether the ingroup members are generally egalitarian or ingroup-favoring when comparing their football teams to other football teams (Do you think [football team name] fans like you are generally egalitarian when comparing their teams to other teams, or do they favor their football team?). Participants answered this question on a scale ranging from 0 (ingroup members always act in accordance with the egalitarianism norm; complete egalitarianism) to 100 (ingroup members always act in accordance with the favoritism norm; complete favoritism). For instance, choosing middle values on this scale may indicate that participants perceive ingroup members as holding mixed or ambivalent attitudes toward resource distribution. That is, they may sometimes act in an egalitarian manner and at other times favor ingroup interests; however, since the midpoint was not labelled, it is also possible that participants interpreted it as a neutral or uncertain stance. We therefore caution against making strong assumptions based solely on midpoint values and interpreting them with this ambiguity in mind. Participants were asked to allocate some financial resources belonging to the Turkish Football Federation (TFF) between their football team and the other 17 football teams in the Turkish Super League to measure the ingroup bias, using three different items (Standardized Cronbach's α = .98). Participants allocated 72 million Turkish Liras (₺) from advertisements in the first item, 54 million ₺ from broadcasting the matches in the second item, and 36 million ₺ from social media sponsorships in the third item among 18 football teams. The participant's decisions of 4, 3, and 2 million ₺ for each item, respectively, mean that their team will be given an equal share with the other teams. We subtracted these amounts (4, 3, and 2) from

the amount of resources participants allocated to their football team in each item and averaged the three final scores to calculate ingroup bias.

**Procedure.** After obtaining institutional IRB approval, we stated that the TFF would make a policy change in distributing its financial resources to Turkish Super League football teams and shape the new policy according to the opinions of football team fans as a cover story. We announced it on the departmental website with the online link of the study created in Qualtrics. Participants first read the informed consent form, and after giving a written online consent, they filled out the demographic information form. They then completed the ingroup identification scale, answered the perceived ingroup norm question, and allocated TFF's financial resources between their football team and other football teams as an outcome measurement. We asked participants for feedback to assess whether they had understood the true purpose of the study. We concluded the study by informing the participants about the real aim of the study. Moreover, we stated that at the end of the research, they could share the departmental website containing the call of the research with the fans around them if they wished. All data were collected online via the Qualtrics link. The data collection process started on September 16, 2019, and was completed on December 2, 2019. Participants completed the research in an average of 9 minutes. We gave a gift card worth 100 ₺ (≈19 US$) to 3 participants as an incentive.

## Results

To test whether the group norm (W) would moderate the relationship between ingroup identification (X) and ingroup bias (Y), a simple moderation model is employed with 5000 bootstraps. Ingroup identification and perceived ingroup norm were centered. No data cells included in the analysis were empty, and no cells were excluded from the analysis for any statistical reason (outliers, etc.).

The overall model was significant, $F(3,184) = 7.37$, $p < .001$, $R^2 = .11$. The effect of ingroup identification on ingroup bias was significant ($b = 2.48$, $t(184) = 4.06$, $p < .001$); however, the effect of ingroup norm on ingroup bias was not ($= .06$, $t(184) = 1.82$, $p = .070$). Interaction between ingroup identification and ingroup norm was also significant ($b = .04$, $t(184) = 1.98$, $p = .049$, 95% CI =[.0001,.0867], $\Delta R^2 = .02$, $F(1,184) = 3.92$. Simple slopes were calculated at one standard deviation above and below the mean of the moderator variable. Higher ingroup identification was associated with higher ingroup bias at the level where the perceived ingroup norm was more ingroup-favoring ($b = 3.59$, $t(184) = 4.18$, $p < .001$); however, there was no effect of ingroup identification on ingroup bias at the level where the perceived ingroup norm was more egalitarian ($b = 1.37$, $t(184) = 1.73$, $p = .086$). On the other hand, at the mean level ($M_{norm} = 69.79$) where the ingroup norm leans

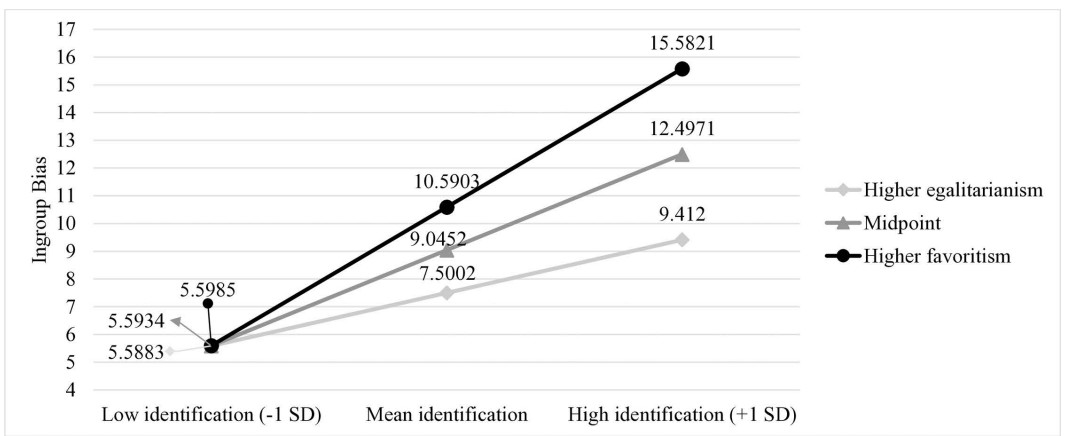

**Fig 1. The moderating role of ingroup norm in Study 1.** The ingroup norm of egalitarianism weakens the association between ingroup identification and ingroup bias.

slightly toward favoritism, higher ingroup identification was associated with higher ingroup bias ($b = .84$, $t(184) = 4.06$, $p = .001$). Here, with the term *more egalitarian*, it is emphasized that the respondents have preferences approximating to the value *0*, which amounts to complete egalitarianism. Fig 1 presents the interaction effect. Levels of the variable were determined according to +1 SD and −1 SD of the mean in Fig 1.

The significant regression coefficient of 3.59 million ₺ shows that when participants perceived their ingroup norm was more ingroup-favoring, each unit of increase in ingroup identification was accompanied by an additional allocation of approximately 350,000 € to their own football team. For interaction effect, post-hoc achieved power analysis using *pwr2ppl* [50,51] revealed that we achieved .814 power.

## Discussion

This cross-sectional study showed that ingroup identification positively predicted ingroup bias while ingroup norms unexpectedly did not. The link between identification and ingroup bias was strengthened when the ingroup norm is favoritism (H2_a), while the significance of this link disappeared in the presence of an egalitarian ingroup norm (H2_b); however, our initial expectation was that under an egalitarian ingroup norm, the relationship between identification and ingroup bias would be negative. Theoretically, based on the relevant literature we had expected that higher identification would predict lower ingroup bias under the egalitarian ingroup norm. However, this hypothesis (H2_b) was not supported [13].

The pattern of stronger adherence to ingroup norm displayed by people with high ingroup identification seems to be very sensitive to the content of the norm. It is argued that egalitarian attitudes *within* groups might have been preceded by fierce competition for resources *between* groups in our evolutionary history [52]. The perception of any kind of competition may orient individuals toward discriminating against outgroups, and when the ingroup norm is favoritism, it seems to exaggerate an already-existing tendency toward ingroup bias. Therefore, the inability or unwillingness to apply egalitarian norms to outgroups might have stemmed from the fact that egalitarian attitudes have been traditionally limited to, and embedded in, intragroup dynamics [53]. In order to act egalitarian toward outgroups, the context in which the intergroup interactions take place must be deprived of the perception of threat and competition; otherwise, egalitarian motivations are likely to be restricted to ingroup members only [54]. This speaks to the possibility that giving up money might have been perceived as a threat to the ingroup's interest.

However, this failure can also be attributed to another explanation. The reason for participants to act against their very own self-reported ingroup norm in sharing resources may be because they had a hard time imagining the ingroup norm in action and what other group members approve of. Alternatively, they might have wanted to see themselves as more egalitarian than they actually were, since egalitarianism might have been perceived as a more morally admirable virtue than ingroup favoritism [55]. Hence in Study 2, we designed an experimental study to manipulate the ingroup norm through a live discussion in a novel way.

## Study 2

In Study 1, self-reported ingroup identification and ingroup norm were only cross-sectionally measured. Although the overall model of moderation was significant, we did not find the effect of egalitarian ingroup norm on the association between identification and ingroup bias. This finding along with the work of Jetten and colleagues [14] and Çoksan and Cingoz Ulu [13], who had not found it either. However, we considered the possibility that highly identified individuals under egalitarian ingroup norms might still exhibit less ingroup favoritism in certain contexts.

To investigate this, we employed a novel methodology of manipulating ingroup norms through a group discussion. While our research question did not concern the differential effects of descriptive and injunctive norms, tapping into the latter was necessary for norm manipulation. This is because injunctive norms, representing what ought to be done, can exert a significant and consistent influence on emotional and behavioral responses toward rival outgroups, whereas the effects of descriptive norms, representing common behaviors, tend to be less consistent and reliable [56]. Therefore, injunctive

norms might be more relevant to group identity than descriptive ones [57]. When individuals perceive an injunctive norm within their ingroup, they experience greater normative pressure to align their behavior accordingly, as failing to do so may threaten their standing within the group [45,58]. This distinction is crucial, as the belief that people should allocate equal resources between ingroup and outgroup members is deemed as an injunctive norm rather than a descriptive one [59]. This means that while people may endorse intergroup equality as the morally right thing to do, they do not necessarily expect it to be widely practiced. This approach aligns with the conceptualization of injunctive norms in earlier work [58,60], where social expectations are communicated through consensus and perceived obligation to the ingroup rather than mere behavioral trends observed within the ingroup.

Hence, simulated group discussion may stress the importance of shared ingroup values and beliefs, thereby promoting more injunctive ingroup norms than descriptive ones. To that end, manipulating ingroup norms through intragroup communication may present a robust way to make it more group-relevant. That is, unlike previous studies [13], we aimed to reveal how group norms are constructed not only through individual perceptions of their prevalence, but also through a simulated process of group discussions. Thus, by providing a more comprehensive understanding of how group norms are dynamically constructed, we aimed to provide an alternative perspective to the unidirectional norm manipulations in the literature. Ingroup identification was also manipulated to clearly establish the causal links in the same moderation model employed in Study 1.

## Method

**Participants.** A hundred and fifty-nine lay people participated in the study recruited via snowball sampling. According to the attention check questions (What was your similarity with the other ingroup members? High, medium, low), 11 participants could not correctly identify the ingroup identification condition they were assigned to. Moreover, five participants could not correctly determine the ingroup norm condition they were assigned to (Which norm do your team fans behave according to your conversation with the other three fans? Egalitarianism, favoritism). In addition, eight participants did not read the statements according to the attention check question (Please mark the answer to this statement as *disagree*. This statement is shown to test whether you have read the questions). According to the feedback we received at the end of the research, one participant identified the real purpose of the research (What do you think is the purpose of the research? Participant's answer: The research aims to examine the relationship between group commitment and the perception of equality). Hence, a total of 25 participants were excluded. The average age of the remaining 134 participants (34 female, 100 male) is 26.28 ($SD = 8.10$). We focused on the fans of the four football teams that won the most championships in Türkiye since the fans of other football teams were very few in the first study. Hence, they are either Fenerbahçe ($N = 54$, 40.3%), Galatasaray ($N = 44$, 32.8%), Beşiktaş ($N = 32$, 23.9%), or Trabzonspor ($N = 4$, 3%) football club fans and, they mostly classify themselves as middle socioeconomic level ($N = 69$, 51.5%).

## Measurements and procedure.

As in the first study, the cover story of the research was to get the fans' opinions for the financial aid policy change that the TFF would distribute to the teams. Participants who read the aim of the research on the departmental website made an appointment with the researchers. They came to the lab in the department at their appointment times. Research assistants who were unaware of the content and the hypothesis of the study followed the participant greeting procedure and seated the participant at the computer (double-blind procedure). After reading the informed consent form and giving written online consent, the participants filled out the demographic information form. It was stated that we would ask various questions to the participants to create their fan-type profiles, and they should answer these questions as quickly as possible to manipulate participants' ingroup identification level. A total of 16 questions (e.g., When was your team founded? Which player on your team was caught offside the most?) were asked randomly from a pool of 36 questions, and the participants were asked to answer these questions as quickly as possible. In addition, the participants evaluated

their teams in terms of values such as honesty and fair play. Participants randomly assigned to the high ingroup identification condition were informed that, based on their survey responses, they were very similar to their team's other fans and were a typical member of their ingroup. Participants randomly assigned to the low ingroup identification condition were informed that, based on prior survey responses, their views and preferences differed from those of the majority of their team's supporters, making them an atypical ingroup member. This manipulation task was adapted from Jetten and colleagues [14]. They were also asked about their similarity level again to understand whether the participants read the information provided. For the manipulation check, they completed the tailored ingroup identification items of Leach and colleagues [48], which we used in the first study. The scale consists of 14 items, and a higher score indicates stronger ingroup identification (Standardized Cronbach's α = .95). An independent t-test revealed that participants assigned to high ingroup identification condition have greater ingroup identification scores ($N=73$, $M=5.22$, $SD=1.19$) than those assigned to low ingroup identification condition ($N=61$, $M=4.66$, $SD=1.40$; $t(132) = 2.51$, $p=.013$), which highlights that the manipulation has achieved its purpose.

Following the manipulation check, to manipulate participants' ingroup norm, they were informed that they would have a written chat with three football team fans who were located at a different cubic in the same lab to discuss the distribution policies, and they entered the chat room by choosing a nickname. There was a moderator and three fans in the chat, but these were actually chatbots preprogrammed by the researchers. The moderator stated that they would ask the participants a question in random order. The participant was told that they remained in the last place. Then the moderator started asking questions that the researchers had determined. Participants randomly assigned to the egalitarianism norm condition read that the other three fans, one at a time and displaying agreement with the previous speaker, advocated for equal distribution of advertising revenue. Participants randomly assigned to the favoritism condition, on the other hand, read that the other three fans advocated for greater allocation of advertising revenues to their own football teams. Participants were explicitly asked to respond in the chat, after viewing others' comments, which encouraged them to actively formulate their own position regarding what ingroup members should support. Although our manipulation draws on both descriptive and injunctive aspects, it was framed and received by participants in a way that repetitively encouraged reflection on appropriate rather than typical behavior. As we stated above, all chat procedure (see Appendix I) was predetermined and automatically run through Java Script coding in the Qualtrics interface.

As an attention check, the participants were asked again what the ingroup norm was according to the chat. For manipulation check, similar to the perceived ingroup norm measurement in the first study, the participants answered two questions (Do you think [football team name] fans like you act egalitarian or ingroup-favoring when allocating TFF advertising revenues?; As a [football team name] fan, are you and other fans generally egalitarian or ingroup-favoring when allocating TFF advertising revenues?; $r=.57$, $p<.001$). An independent t-test revealed that participants assigned to egalitarianism ingroup norm condition perceived their ingroup norm as more egalitarian ($N=72$, $M=34.87$, $SD=31.21$) than those assigned to favoritism ingroup norm condition ($N=62$, $M=65.50$, $SD=23.80$; $t(132) = 6.31$, $p<.001$), which highlights that the manipulation has achieved its purpose. The outcome measurement was the same as in the first study (three items, Standardized Cronbach's α = .93).

We received feedback from the participants to see if they understood the real aim of the research and concluded the study by informing the participants about the real aim of the study. As in Study 1, we also highlighted that participants could share the department website containing the call text of the research with the fans around them if they wished. Each participant was admitted to the lab individually and did not communicate with anyone throughout the experiment. Participants left the lab after informing the research assistant accompanying them that they were done, and thus the experiment was completed.

We started collecting data on March 8, 2021, and the data collection process ended on July 2, 2021. The time gap between the two studies resulted from the COVID-19 pandemic and the fact that this study required a lab setting.

Participants completed the research in an average of 30 minutes. We gave a gift card worth 100 ₺ to 3 participants as an incentive.

## Results

We ran a 2 X 2 between-subjects ANOVA, which revealed that both the effect of ingroup identification ($F(1,130) = 4.66$, $p = .033$, partial $\eta^2 = .035$) and ingroup norm ($F(1,130) = 23.09$, $p < .001$, partial $\eta^2 = .151$) on ingroup bias were significant. Participants with high ingroup identification showed more ingroup bias ($M = 5.60$, $SE = .60$) than those with low ingroup identification ($M = 3.68$, $SE = .65$), and participants with favoritism ingroup norm showed more ingroup bias ($M = 6.77$, $SE = .65$) than those with egalitarianism norm condition ($M = 2.51$, $SE = .61$).

Moreover, although the interaction effect was not statistically significant ($F(1,130) = 1.83$, $p = .177$, partial $\eta^2 = .014$), we conducted pre-planned post-hoc comparisons to further examine the hypothesized moderation effect (H1, H2$_a$, and H2$_b$), in line with recommendations in the literature suggesting that such comparisons may still be warranted when the interaction is theoretically important or expected. Post-hoc comparisons indicated that, among participants in the low ingroup identification condition, there was more ingroup bias in the favoritism norm condition ($N = 30$, $M = 5.21$, $SE = .93$) compared to the egalitarianism norm condition ($N = 31$, $M = 2.15$, $SE = .92$; $p = .021$, 95% CI [.476, 5.645]). Similarly, among participants in the high ingroup identification condition, those in the favoritism norm condition ($N = 32$, $M = 8.33$, $SE = .90$) displayed greater ingroup bias compared to participants in the egalitarianism norm condition ($N = 41$, $M = 2.86$, $SE = .80$; $p < .001$, 95% CI [3.092, 7.852]). In the favoritism norm condition, participants with high ingroup identification ($M = 8.33$, $SE = .90$) showed more ingroup bias compared to those with low ingroup identification ($M = 5.21$, $SE = .93$; $p = .017$, 95% CI [.558, 5.686]). However, in the egalitarianism norm condition, there was no significant difference in ingroup bias between participants with high and low ingroup identification ($p = .559$, 95% CI [−1.690, 3.113]). Results are summarized in Fig 2.

To provide convergent validity and methodological triangulation of our experimental findings, we additionally conducted a supplementary moderation analysis using participants' continuous scores on the manipulation checks of perceived ingroup identification and perceived ingroup norms. It is important to clarify that although these measures initially served as manipulation checks, employing them in a moderation model allows us to examine the robustness of our experimental

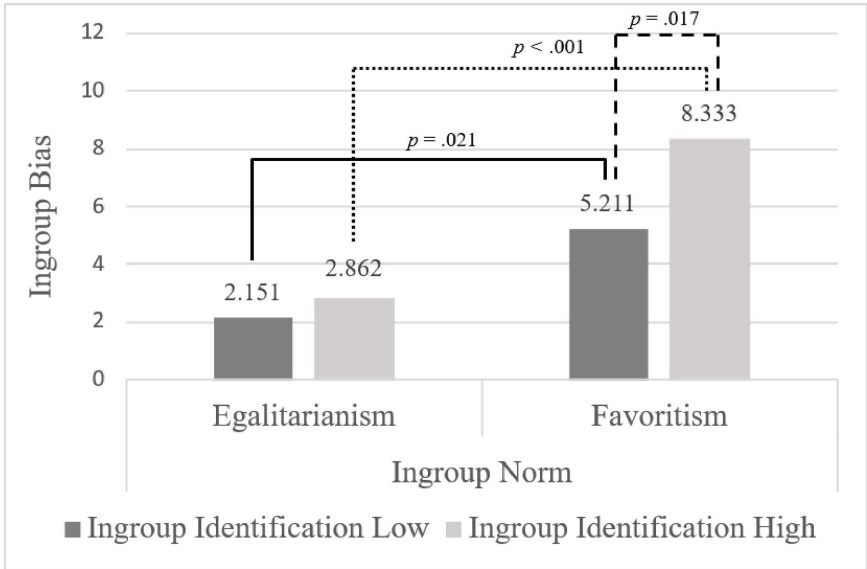

**Fig 2. Group comparisons between experimental conditions.** Intergroup comparisons point to a possible moderating role of group norm.

results through an alternative analytic strategy [61]. Specifically, while the ANOVA approach tested the causal impact of manipulated categorical variables, this complementary moderation analysis explores how individual differences in response to manipulations may be systematically related to ingroup bias. This supplementary analysis was preplanned and serves to support the primary experimental analyses rather than to independently test our hypotheses. Moreover, the moderation model conducted in Study 2 parallels the analytical approach employed in Study 1. Thus, presenting this analysis allows for direct methodological consistency and comparability across studies. Specifically, it enables us to assess whether the moderation observed through measured continuous variables in Study 1 replicates conceptually under experimental manipulation conditions in Study 2, thereby offering a more integrative and robust narrative across studies.

Hence, we ran a simple moderation model with 10000 bootstraps to examine our hypotheses. Ingroup identification and perceived ingroup norm, measured as continuous variables, were centered. The overall model was significant, $F(3,130)$ = 31.57, $p < .001$, $R^2 = .42$. The effect of ingroup identification ($b = .84$, $t(130) = 2.85$, $p = .005$) and ingroup norm ($b = .10$, $t(130) = 7.97$, $p < .001$) on ingroup bias, as well as the interaction between them were significant ($b = .02$, $t(130) = 2.36$, $p = .017$, 95% CI = [.0038,.0373], $\Delta R^2 = .03$, $F(1,130) = 5.87$. Simple slopes were calculated at one standard deviation above and below the mean of the moderator variable. As in the first study, higher ingroup identification was associated with higher ingroup bias at the level where the perceived ingroup norm is more ingroup-favoring ($b = 1.49$, $t(130) = 2.85$, $p = .001$); however, there was no effect of ingroup identification on ingroup bias at the level where the perceived ingroup norm was more egalitarian ($b = .19$, $t(130) = .52$, $p = .607$). Moreover, at the mean level ($M_{norm} = 49.06$) where the ingroup norm clearly indicates neither egalitarianism (0 = perceived ingroup norm is complete egalitarianism) nor favoritism (100 = perceived ingroup norm is complete favoritism), higher ingroup identification was associated with higher ingroup bias ($b = .84$, $t(130) = 2.85$, $p = .005$). The interaction effect of ingroup identification and perceived ingroup norm is presented in Fig 3 where levels of the variable were determined according to +1 SD and −1 SD of the mean.

The significant regression coefficient of 1.49 million ₺ shows that when participants perceived their ingroup norm is more ingroup-favoring, each unit of increase in ingroup identification was accompanied by an additional allocation of approximately 145,000 € to their ingroup. Post-hoc power analysis using *pwr2ppl* [50,51] showed that we achieved .842 power for the interaction effect.

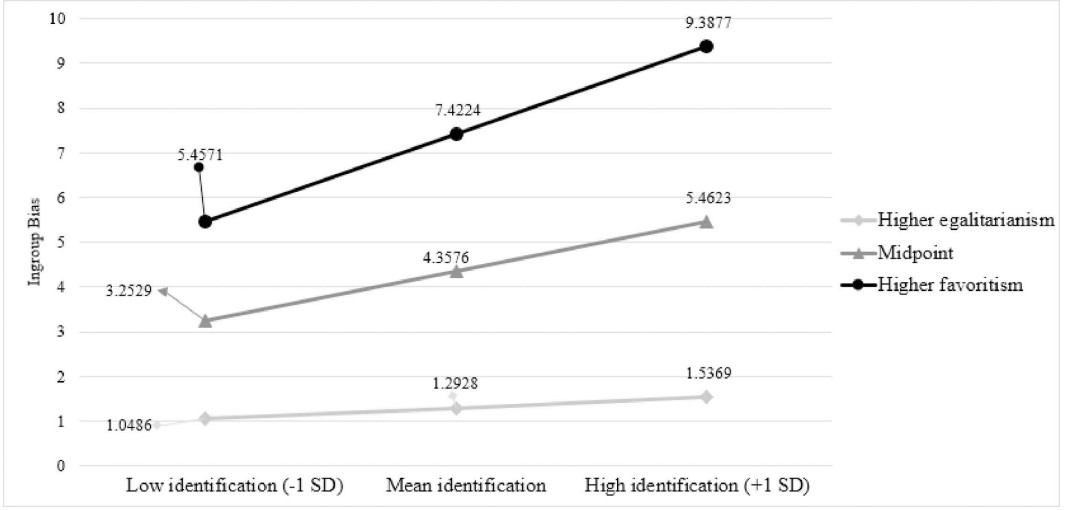

**Fig 3. The moderating role of ingroup norm in Study 2.** The ingroup norm of egalitarianism weakens the association between ingroup identification and ingroup bias in Study 2.

## Discussion

Study 2 extended the results of Study 1. In parallel with Study 1, ingroup identification positively predicted favoritism. However, unlike Study 1, we observed a main effect of the norm; participants showed more ingroup bias when they reported that ingroup norms were more indicative of favoritism. Moreover, under the favoritism norm condition, people with high ingroup identification appeared to be more influenced by the ingroup norm than people with low ingroup identification. In short, under the favoritism norm condition, people with high ingroup identification showed more favoritism, whereas when the ingroup norm indicated egalitarianism, as in the first study, the relationship between ingroup identification and ingroup bias lost statistical significance ($H2_b$).

It is important to note that while the ingroup norm did not have a main effect on ingroup bias in Study 1, it exerted a significant influence in Study 2. This suggests that our manipulation might have amplified the effect of ingroup norms. However, as the effect of social norms was also observed when analyzing measured norms, we recognize that norm salience likely contributed to the observed effects rather than being their sole cause. Intragroup discussions may prompt individuals to reflect on their place in the group, and this enhances one's commitment to their ingroup's shared opinions [62]. As with Study 1, however, we failed to find evidence to support our hypothesis ($H2_b$) of a negative identification-ingroup bias association on egalitarianism norm condition. It seems to be the case that the challenges in extending egalitarian treatment to outgroups discussed in the first study were also upheld by the results of Study 2. Importantly, the results of the ANOVA and the simple moderation analysis are consistent. Although our related hypothesis was not supported, both analyses showed that participants with high ingroup identification exhibited more ingroup bias when the ingroup norm promoted favoritism, whereas this significant relationship was not observed under the egalitarian norm condition.

## General discussion

Our main expectation that those with higher identification in the egalitarian norm would favor their ingroup to a lesser extent was not supported in either of the studies ($H2_b$) although our other findings (regarding H1 and $H2_a$) seem to be parallel with the social identity literature. According to this approach, individuals who strongly identify with their group are motivated to be more responsive to group norms and to maintain in-group cohesion. They make more effort to maintain their commitment to the group and to gain social acceptance by conforming to group norms. Therefore, when ingroup norms emphasize egalitarianism, individuals with high ingroup identification were expected to exhibit more egalitarian behavior. However, the ingroup bias of members under the egalitarianism norm did not differ depending on their ingroup identification. Likewise, a similar hypothesis was developed in previous studies [13,14], but their hypotheses were not supported. Unlike previous studies, we examined the effect of the norm of egalitarianism instead of fairness, and manipulated the group norm with a group discussion.

Our findings showed that the ingroup norm moderates the ingroup identification–ingroup bias association among Turkish football fans, who represent real-life groups engaged in competitive but not overtly conflictual intergroup relations. This extends previous research, which has examined this moderating effect in both high-conflict contexts [13] and in settings involving minimal group distinctions with no real-world stakes [14].

Moreover, our results demonstrated that while the favoritism norm led to an increase in ingroup bias, the egalitarian norm reduced ingroup bias only in Study 2. This finding suggests that in Study 1, self-reported ingroup norms were not significant predictors of ingroup bias, whereas in Study 2, the experimentally induced ingroup norm had a more pronounced predictive effect. We may attribute this outcome to the specific methodology employed for norm manipulation [18]. Accordingly, we propose that the simulated group discussion in chat rooms may offer a viable and effective approach for altering ingroup norms. This method's effectiveness aligns with Moscovici's [43] theoretical perspective on injunctive group norms and social psychological research emphasizing that group norms and their behavioral influence emerge more robustly through social interactions rather than self-reported measures or unilateral transmission of descriptive norms [59].

Furthermore, although ingroup norm manipulation in Study 2 showed a more pronounced effect on ingroup bias than perceived ingroup norm in Study 1, once again, our key expectation of a lower ingroup bias among people with high ingroup identification under egalitarianism norm was not supported (H2$_b$). The lack of significant differences in ingroup bias between people with high and low ingroup identification under egalitarian norm implies that normative influence in shaping behavior is contingent on the content of the norm. We concur with Jetten and colleagues [14] (p. 608) that the egalitarianism norm, compared to the ingroup favoritism norm, is "*inherently less group-relevant and group-defining.*" The greater responsiveness of group members to favoritism over egalitarian norms, regardless of identification level, may stem from the perceived cost of adhering to egalitarianism norm. To explain this cost, we must first refer to a crucial finding: Falomir-Pichastor and colleagues [21] found that an ingroup norm of anti-discrimination concerning the treatment of foreigners (both conceptually and operationally akin to egalitarianism in our research) was perceived as a threat to ingroup's privileged position, particularly by people with high ingroup identification. The authors argue that strongly identified individuals find themselves to be torn between complying with the egalitarian ingroup norm and looking out for the ingroup's interest. Out of this conflict, the latter motivation comes out victorious on account of its lower perceived cost. The strategy with which they compensate for weakening their commitment to (egalitarian) ingroup norm is to strengthen their ties to ingroup in other domains, such as ingroup values and ingroup identification. Therefore, people with high ingroup identification may become more invested in their identity to atone for not aligning themselves with the egalitarian ingroup norm [15].

Finally, to reiterate, the ANOVA results in Study 2 are consistent with the findings from the simple moderation analysis. In the primary interaction-focused results, participants with high ingroup identification displayed greater ingroup bias compared to those with low identification, and participants under the favoritism norm condition exhibited more ingroup bias than those under the egalitarian norm condition. Moreover, within the favoritism norm condition, those with high ingroup identification showed more bias in favor of their ingroup than those with low identification. This pattern may suggest, as previous research has indicated, that a favoritism norm legitimizes bias by setting an expectation to prioritize ingroup interests and that this norm maximizes ingroup investment, especially among those with high identification. The finding that a favoritism norm triggers bias among highly identified individuals may support the view that "identity-congruent norms" amplify this behavior [63]. However, the fact that the egalitarian norm dampens this effect suggests that normative pressure can moderate the influence of identification. Overall, our findings indicate that ingroup bias is shaped not only by individual identity processes but also by collective group norms, thereby aligning with multilayered approaches proposed in the literature [64].

On the other hand, there are certain limitations of our research. Firstly, our manipulation did not include direct group discussion in the form of back-and-forth norm negotiation. Standardizing the manipulation required avoiding a back-and-forth interaction, as ensuring standard responses across various inputs from participants would be unfeasible. However, consistent with earlier work [43,45,58], our design still captured a key mechanism of norm formation, via group consensus and social exposure,. This was achieved by structuring sequential ingroup member statements regarding appropriate behaviors, and thus fostering the perception of a normative expectation, more closely aligning with injunctive than descriptive norms [59]. Although group identification and perceived group consensus may be sufficient to infer and act on norms even in the absence of direct negotiation [65], future research may incorporate AI-driven, adaptive chat interfaces where participants engage in dynamic back-and-forth discussions with programmed ingroup members. This would allow for a standardized but simulated norm negotiation process, addressing the limitation of the current design while maintaining experimental control.

Second, it is important to acknowledge the limitations of our sample. Due to practical constraints on representative sampling, we collected data from fans of the most affluent and well-established Turkish football teams. Therefore, it would be reasonable to consider our participants as fans of advantaged groups or even advantaged groups themselves. This raises the possibility that participants may have endorsed the idea of allocating more resources to teams with larger fan

bases, as this unequal distribution aligns with the current status quo, and may be perceived as fair. The perceived fairness of getting the biggest portion of the budget for themselves can be in part responsible for setting favoritism norm as default [37]. Nevertheless, if participants are prompted to consider the more pressing needs of smaller teams, the norm could exhibit a stronger shift toward egalitarian treatment of others [66]. This implies that a focus on the differential needs of advantaged and disadvantaged groups can amplify the effect of egalitarianism norm in counteracting the tendency for highly identified individuals to unconditionally prioritize (advantaged) ingroup interests.

Third, although power analyses for the simple moderation models indicated that we had sufficient statistical power, our statistical power for detecting the interaction effect in Study 2's ANOVA appears to be quite low (see Appendix II). Future studies may address this limitation by increasing sample sizes.

Fourth, we acknowledge that our sample consisted of self-identified football fans whose level of attachment to their teams may have varied. While our manipulation successfully altered perceived ingroup identification, we cannot fully account for participants' baseline levels of team commitment. Moreover, we recognize that in some real-world contexts, relationships between fans of different teams can involve heightened intergroup tension. In such settings, ingroup norms may function differently, as baseline group commitment and intergroup hostility could interact with normative influences.

Moreover, the presence of larger, shared goals can also deflate ingroup bias. As McGuire and colleagues [17] demonstrated, individuals may prioritize egalitarianism over ingroup favoritism when there is a larger and shared goal for all groups, such as a more intense future competition. In such cases, individuals may be more inclined to provide equal opportunities to every group in the current competition in order to maximize the chances of achieving the shared goal. Future research could investigate whether the combination of egalitarian ingroup norms with shared goals can reduce ingroup bias among highly identified group members.

In conclusion, egalitarianism norm can serve as a vital factor to mitigate ingroup favoritism, in support of Çoksan and Cingöz Ulu [13]. Although our findings did not show that people with high ingroup identification under the egalitarianism norm favor their ingroup less than people with low ingroup identification, unlike the situation in the favoritism norm condition, having the norm of egalitarianism successfully reduces ingroup favoring in those with higher identification. Based on these findings, which were repeated across two studies, promoting egalitarian ingroup norms and values can reduce ingroup bias and, accordingly, outgroup discrimination, albeit speculatively. To foster a more cohesive society, it is crucial to instill egalitarian values in social institutions to counteract the deep-seated tendency towards ingroup favoritism.

## Acknowledgments

We would like to thank Cansu Yumuşak, MSc, for helping us during the data collection.

## Author contributions

**Conceptualization:** Sami Çoksan, Ahmed Faruk Sağlamöz.

**Data curation:** Sami Çoksan, Ahmed Faruk Sağlamöz.

**Formal analysis:** Sami Çoksan.

**Investigation:** Sami Çoksan, Ahmed Faruk Sağlamöz.

**Methodology:** Sami Çoksan.

**Project administration:** Sami Çoksan.

**Resources:** Sami Çoksan.

**Visualization:** Sami Çoksan.

**Writing – original draft:** Sami Çoksan, Ahmed Faruk Sağlamöz.

**Writing – review & editing:** Sami Çoksan, Ahmed Faruk Sağlamöz.

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
