## [Decision Letter · Decision Letter 0]

11 Feb 2025

PONE-D-24-50337
The ingroup norm of egalitarianism eliminates the association between ingroup identification and ingroup bias
PLOS ONE

Dear Dr. Çoksan,

Thank you for submitting your manuscript to PLOS ONE. After careful consideration, we feel that it has merit but does not fully meet PLOS ONE’s publication criteria as it currently stands. Therefore, we invite you to submit a revised version of the manuscript that addresses the points raised during the review process.

Both reviewers see some merits in your manuscript but also found the current form of the manuscript needs major revisions. Both reviewers offered constructive comments from how to restructure the introduction section and the hypotheses section. Both reviewers also have suggestions for how the detailed information of the studies could be provided to assist the readers in understanding the design of the studies. Both reviewers also offer suggestions for the results section. At the current form of the manuscript, it is difficult to evaluate whether the manuscript would be further processed. However, it seems that all of the major concerns could be addressed by major revision. Thus, I invite you to submit a revised manuscript.

We look forward to receiving your revised manuscript.

Kind regards,

I-Ching Lee

Academic Editor

PLOS ONE

Journal requirements:  
 
When submitting your revision, we need you to address these additional requirements.
 
1. Please ensure that your manuscript meets PLOS ONE's style requirements, including those for file naming. The PLOS ONE style templates can be found at 
https://journals.plos.org/plosone/s/file?id=wjVg/PLOSOne_formatting_sample_main_body.pdf and 
https://journals.plos.org/plosone/s/file?id=ba62/PLOSOne_formatting_sample_title_authors_affiliations.pdf.
 
2. Please amend either the title on the online submission form (via Edit Submission) or the title in the manuscript so that they are identical.
 
3. Please include your full ethics statement in the ‘Methods’ section of your manuscript file. In your statement, please include the full name of the IRB or ethics committee who approved or waived your study, as well as whether or not you obtained informed written or verbal consent. If consent was waived for your study, please include this information in your statement as well. 

Reviewers' comments:

Reviewer's Responses to Questions

**Comments to the Author**

1. Is the manuscript technically sound, and do the data support the conclusions?

Reviewer #1: Partly

Reviewer #2: Partly

2. Has the statistical analysis been performed appropriately and rigorously? 

Reviewer #1: Yes

Reviewer #2: Yes

3. Have the authors made all data underlying the findings in their manuscript fully available?

Reviewer #1: Yes

Reviewer #2: Yes

4. Is the manuscript presented in an intelligible fashion and written in standard English?

Reviewer #1: Yes

Reviewer #2: Yes

5. Review Comments to the Author

Reviewer #1: In two studies (one cross-sectional and one experimental), conducted in the intergroup context of fans of Turkish football teams, it was tested whether social norms (favouring the ingroup vs. egalitarian) moderate the association between ingroup identification and ingroup bias (operationalized as [un]equal distribution of financial resources across football clubs). It was found that ingroup identification was associated with ingroup bias only when norms favour the ingroup, while there was no association between ingroup identification and ingroup bias when norms were egalitarian.

The research question and the hypothesis were widely explored by previous research, and findings are fully in line with previous research. However, replication of well-established findings is important, and the research program includes two novelties, i.e. the intergroup context (relationships between fans of Turkish football teams) and a novel experimental manipulation of social norms.

Despite these positive remarks, in my view the article should be thoroughly revised, because the organization of the introduction and the presentation of some findings is suboptimal and not fully clear.

Introduction and theoretical justification of the research program

1. At least two issues seem crucial for the novelty of the research program, but are undertheorized. First, given that a lot of attention is paid to the importance of focusing on egalitarian norms instead of fairness norms, authors should elaborate more on the theoretical distinction between fairness and egalitarianism and, if existent, present research showing different outcomes between fairness and egalitarianism social norms, ideologies, and so on. Second, I would encourage the authors to explain more thoroughly why they think that relationships between fans of different football teams are not conflictual. News often reports clashes between rival fans, and this appears to have happened recently also in Turkey. Therefore, I do not find convincing the argument that the authors have considered a non-conflictual intergroup context. Furthermore please note that, on page 13, I would expect an explanation about the differential effects of descriptive and injunctive norms. Otherwise, the claim that injunctive norms might be more relevant to group identity than descriptive norms seems not well justified.

2. The authors have summarized literature suggesting that social norms are also formed and negotiated via social interactions in group discussions, emphasizing the active role of each group member. While I find the arguments convincing, and at the same time I have appreciated the novel and creative experimental manipulation of social norms, built on classic research, I think there is a mismatch between the theoretical rationale and the actual experimental manipulation. Indeed, there were no real group discussions, and therefore there was no negotiation. Each participants was merely invited to express their opinion after being exposed to the opinion of three other alleged ingroup members. I would encourage the authors to carefully think about the best justification for their method, and to avoid overemphasizing some strengths which might not fully apply to their method.

3. The organization of the introduction is not optimal. I would suggest to first summarize relevant literature, and then state specific hypotheses which should be based on literature. As a minor related issue, I find it surprising to first have the hypotheses about the interaction effects and then the hypotheses of the main effects. The reverse order sounds more logical. Most importantly, the section “current research” is extremely long and not very focused. For example, the first page of “current research" should be part of the theoretical summary. Most of the paragraphs are about the conceptualization of social norms (favouring the ingroup vs. egalitarian) and on the novel experimental manipulation of social norms. However, in “current research” I would expect paragraphs explaining and justifying the whole research program, and not a focus mainly on the second study. Overall, the organization of the whole introduction (from the very first page to “current research”) should be improved.

Studies 1 and 2

4. The discussion of Study 1 is partly confusing. Authors state “Moreover, high identifiers showed higher levels of ingroup bias under the favouritism norm (H1 & H2).” However, H1 and H2 are presented as hypothesis about the effects of ingroup identification for those who perceive norms as favouring the ingroup (H1) or egalitarian (H2) and not as a single hypothesis that egalitarian norms buffer the effect of ingroup identification. Then H2 is not confirmed, and the above-mentioned sentence is very imprecise.

5. The presentation of Study 2 findings is very confusing. Authors first present findings from the ANOVA, and this is fine given the 2 X 2 experimental design. It is also fine to me to decompose the interaction even if it was not significant. However, why decomposing the interaction also with social norms as predictor and ingroup identification as moderator (“Post-hoc comparisons between experimental conditions indicate that participants with favouritism ingroup norm show more ingroup bias (N = 30, M = 5.21, SE = .93) than those with egalitarianism norm condition (N = 31, M = 2.15, SE = .92; p = .021, 95% CI [.476, 5.645]) on low ingroup identification condition. Similarly, participants with favouritism ingroup norm show more ingroup bias (N = 32, M = 8.33, SE = .90) than those with egalitarianism norm condition (N = 41, M = 2.86, SE = .80; p < .001, 95% CI [3.092, 7.852]) on high ingroup identification condition.”)?

Most importantly, the following section is extremely confusing. What have the authors exactly done? Have they tested the same regression analysis of Study 1 with the measured variables (and not with the experimental manipulations)? If yes, this needs to be very well specified. The sentence “Since 2 X 2 between-subjects ANOVA results indicate the ingroup norm’s moderating role, …” is not a sufficient justification. Importantly, it makes no sense to calculate power on the explained variance of the regression analysis with measured variables. The main novelty of Study 2 is the experimental design, therefore power needs to be calculated based on ANOVA findings (or alternatively on a regression analysis with the two experimental manipulations and their interaction as predictors).

6. The discussion of Study 2 is very confusing. First, the wording “high identifiers” and “low identifiers” should not be used if authors are referring to “participants in the high identification condition” and “participants in the low identification condition”. Second, the sentence “High identifiers were more strongly influenced by ingroup norm than low identifiers under the favouritism norm condition” is very imprecise. In both identification conditions, those in the favouritism norm condition reported higher intergroup bias than those in the egalitarian norm condition, and the interaction was not significant. The authors state “It is essential to note that the ingroup norm did not have a main effect on ingroup bias in Study 1, while it produced a potent effect in Study 2, especially on those in the high identification condition. This must be the direct result of the manipulation method exclusively designed for this study”. However, their justification is in my view not correct. Indeed, in Study 2 the main effect of social norms is detected also when analysing measured social norms (and not only experimentally manipulated social norms).

7. Authors should report in the main text a justification for the sample size of both Studies. Please note that in experimental study 2 there are less than 50 participants per cell, therefore the study might be underpowered (see also the fifth comment).

Discussion

8. The discussion should be thoroughly revised after modifications are implemented to the introduction and to the presentation of Study 1 and 2 findings. Indeed, in the current version of the discussion, there are imprecisions. For example, the sentence “Moreover, the variance of ingroup bias explained by ingroup identification, ingroup norm, and their interaction is 11% in Study 1, while it is 42% in Study 2” should not be used as an illustration of the effectiveness of the social norms experimental manipulation, because it appears to refer to the regression analysis with measured variables in both studies. To make a proper comparison, authors should check the Study 2 variance explained in the ANOVA. As another example, on page 23, line 511, the authors discuss the role of “social interactions” in their social norms experimental manipulation. However, there were no social interactions, as participants expressed their opinion after being exposed to the opinion of alleged ingroup members.

Finally, I would recommend the authors to carefully re-check the article before resubmission.

I hope that these comments will be useful to the authors for their ongoing and future research.

Reviewer #2: The paper investigates the effect of group norms on ingroup bias, in particular, via comparing the induced norm of egalitarian distribution across groups to the induced norm of ingroup favoritism. In Study 1, participants themselves state the strength of their group identification, whereas in Study 2, this is manipulated in the laboratory by random assignment. The study relies on deception to generate the strength of the level of ingroup identification and different group norms.

My main comments concern hypotheses construction, the description of the design, the interpretation of the design and results of both studies.

Hypotheses:

The authors state the following: "Throughout the two studies in the current research, we hypothesized that (H1) when the

ingroup norm is favouritism, participants show more ingroup bias as ingroup identification increases, and (H2) when the ingroup norm is egalitarianism, participants show less ingroup bias as ingroup identification increases since highly identified group members are more inclined to conform to perceived ingroup norms. In terms of main effects, our expectation is

that (H3) participants with high ingroup identification show more ingroup bias than participants with low ingroup identification, and (H4) those in the condition where the ingroup norm is favouritism show more ingroup bias than those in the condition where the ingroup norm is egalitarianism."

Here, the second and the third hypothesis is at odds: If the ex-ante expectation is that high group identification leads to less ingroup bias if the group norm is egalitarian, then the main effect cannot be that ingroup bias is larger with high group identification -this will only be driven by the ingroup favoritism group norm.

Subsequent paragraphs on page 5 explain how those with low group identification would be predicted to behave. Here, the writing is unclear, as this prediction only makes sense as a comparison to those with high group identification in the current study, not stand alone. For example, what would this predict for people with low group identification: "these members still tend to look to ingroup norms to determine the correct behaviour in the social environment because these contexts provide strong guidance to members on what they should do. Thus, even individuals with low ingroup identification still follow these strong social cues to determine appropriate behaviour in the social context, indicating a tendency to conform to group norms, though to a lesser extent than high identifiers" Does this mean that those with low group identification have low ingroup bias generally or only lower than those with high group identification. If the former, is it because they are a different type of person (for example, they may be generally less "groupy" as in recent work by Kranton) or is it because, they care less about their one particular group identity.

Design:

The description of the design leaves out too many relevant details. It should be clarified how the snowball sampling was done upfront. Was it only a website, and how can there be so many recruits from a department website only? We learn much later in the paper that the participants are university students, this should be stated early on. Why call them lay people? Are there any persons who are not students? Is the first study a laboratory study -it looks like, it was not-, this is not written anywhere. How were the participants invited to the laboratory in the second study if they were participating one by one? Does this mean that there were 159 sessions? If so, how was this done, and how is this feasible?

Interpretation of Study 1 results:

The authors state the following on page 11: "however, there is no effect of ingroup identification on ingroup bias at the level where the perceived ingroup norm is more egalitarian (b = 1.37, t(184) = 1.73, p = .086)." Here, there is a moderate effect, which is not significant at the conventional level. This points to the study being underpowered. The achieved power calculation in the paper is of no guidance here either, because it is based on the overall model fit, not on the detection of an interaction effect. As such, the interpretation should reflect this, and it cannot be confidently stated that there is no effect, if anything, an egalitarian group norm deflates ingroup bias level of high group identifiers. Relatedly, Figure 1 caption states "The ingroup norm of egalitarianism eliminates the association between ingroup identification and ingroup bias." I find it misleading. egalitarianism norm weakens the association is a better description of their data.

In general, I find it very problematic to name egalitarianism "complete egalitarianism" (is there partial egalitarianism, if so, how does that look like?), and there is nothing neutral about the mid-point in their measure. One can only state that the scale from 0 to 100 measures the degree of ingroup favoritism.

Interpretation of Study 2:

On page 13, last paragraph, the authors discuss that injunctive norms may affect behavior more strongly than descriptive norms. This is likely the case, however, their design has no clear injunctive norm dimension -or if it is there, it is not explained. Moreover, their design inevitably has a descriptive norm component. These paragraphs should be clarified with respect to what the study design entails, and it should not be implied that the discussion generates an injunctive norm.

Figure 2 caption states "Intergroup comparisons point to a possible moderating role of group norm." This caption should only reflect the results, not an interpretation. So either write that group norm moderates ingroup bias, or that ingroup bias is lower with an egaliatarian norm for all group identification levels. Likewise, Figure 3 caption should just describe the result.

Other comments:

After reading the introduction, and current study sections, Study 1 comes as a surprise, because the main contribution statements relate to Study 2. So at first, I was confused that Study 1 did not have a chat.

The Discussion section, first two paragraphs should be rewritten both for clarity and for language use, e.g., "a potent effect", "hardships of applying egalitarianism". The general discussion that immediately follows is also hard to follow, either state what the hypotheses were and whether they received support in the data, or do not mention them at all. Rephrase "our much sought-after expectation". Rephrase "let us not abandon all hope."

In the same paragraph, once Turkey and once Türkiye is written.

6. PLOS authors have the option to publish the peer review history of their article (what does this mean?). If published, this will include your full peer review and any attached files.

Reviewer #1: No

Reviewer #2: No

---

## [Author Response · Author response to Decision Letter 1]

14 Mar 2025

Response Letter

Dear Editor,

We are deeply grateful to you and the reviewers for the insightful and constructive feedback on our paper. We believe that these helped us immensely in improving the manuscript. We reorganized the introduction, method, and discussion sections according to reviewers’ feedback. We made numerous additions to emphasize the context and social identity background of the paper’s rationale. We tried to focus on the purpose of the paper with new headings. We reformulated our hypotheses to address the problems related to the hypotheses raised by both reviewers. We rewrote the entire text according to these changes. We have re-numbered each comment and provided our answers below each. We indicated the changes in red ink. We hope that this version of the paper is suitable for the journal.

Reviewer #1

1. At least two issues seem crucial for the novelty of the research program, but are undertheorized. First, given that a lot of attention is paid to the importance of focusing on egalitarian norms instead of fairness norms, authors should elaborate more on the theoretical distinction between fairness and egalitarianism and, if existent, present research showing different outcomes between fairness and egalitarianism social norms, ideologies, and so on.

Response: We thank Reviewer 1 for their suggestion to further elaborate on the theoretical distinction between fairness and egalitarianism and to incorporate research demonstrating different outcomes associated with these social norms. While there is no empirical work to the best of our knowledge comparing the effects of fairness and egalitarianism norms, we believe that our existing rationale provides a strong foundation for this distinction, particularly in explaining why fairness may not always translate to equal treatment, and how the egalitarian norm offers a more consistent and neutral framework for resource allocation. Specifically, we highlight that fairness is a more self-serving and context-dependent principle, encompassing multiple distributive norms that may or may not include concerns around equality, group status and group needs; whereas egalitarianism is more narrowly defined as a commitment to strict equality in outcomes, showing less sensitivity to differences in status or needs.

That said, we agree that it needs further conceptual clarification. We have now expanded our explanation with resources demonstrating the theoretical distinction, notably through the context-dependent nature of fairness that heavily limits the external validity of earlier findings. We believe these additions enhance the necessity of this distinction and provide a clearer rationale for our focus on egalitarian norms. Hence, we revised the related content under the new heading of Fairness vs. Egalitarianism on pages 7-8 as follows:

The way we operationalize the ingroup norm manipulation differs from Jetten and colleagues [14], who examined norms under fairness and favouritism conditions. However, a key issue with fairness is that it does not inherently promote equality between groups. In fact, fairness may paradoxically reinforce ingroup favouritism when it is perceived as a means to rectify past injustices [31]. Favouring the ingroup can be interpreted as a fair strategy under zero-sum conditions as well, given that even non-zero-sum games are associated with a fiercer competition over resources than they actually are [32]; after all, resource allocation tasks are by default a zero-sum game. Research also suggests that fairness judgments may be anchored in proportionality rather than strict equality [54]. That is, individuals tend to assess fairness based on the relative contributions of parties involved rather than an egalitarian principle that mandates equal division, leading to outcomes where inequality is perceived as fair.

Another factor that can shape fairness perceptions is status concerns, as higher-status group members are more likely to see unequal outcomes as fair when they believe their group is entitled to greater rewards [60]. This highlights that fairness can be self-serving as it enables high-status groups to justify and maintain inequalities while presenting them as fair [33]. Even in the absence of clear status differences, people still apply fairness flexibly to justify unequal distributions. For instance, in ultimatum game scenarios, individuals tend to offer unequal distributions while believing their offers are fair, despite the fact that recipients consider these allocations as unfair [61]. This finding demonstrates how fairness norms can be misaligned between groups, further reinforcing the idea that fairness can be self-serving and does not inherently promote equality.

Given that fairness is an ambiguous concept subject to multiple interpretations -including compensatory justice, proportionality, and status maintenance- it is challenging to ensure that fairness norms consistently lead to non-favouring distributions. This conceptual ambiguity has led researchers to replace fairness with egalitarianism as a more consistent, neutral, unambiguous and non-retaliatory framework for resource allocation [13]. By emphasizing strict equality in outcomes, egalitarian norms rule out context-dependent interpretations of fairness, which can be endorsed to justify ingroup favouritism. Thus, our study chooses the norm of egalitarianism over fairness to avoid the unintended biases that fairness-based distributions can easily introduce. By focusing on an unequivocally equal allocation of resources, we ensure that fairness, when interpreted in a way that permits ingroup favouritism via status reinforcement or retaliatory motives, does not undermine the impartiality of our norm manipulation.

We have also added the following paragraph on pages 26-27 to emphasize that this area is open to future research, noting the lack of studies directly comparing the effects of fairness and egalitarianism across different social contexts.

Third, we acknowledge that further empirical studies are needed to systematically compare the effects of fairness and egalitarianism norms across different intergroup settings. Future research should explore whether fairness norms lead to different intergroup outcomes compared to egalitarianism, especially in competitive versus cooperative contexts.

2. Second, I would encourage the authors to explain more thoroughly why they think that relationships between fans of different football teams are not conflictual. News often reports clashes between rival fans, and this appears to have happened recently also in Turkey. Therefore, I do not find convincing the argument that the authors have considered a non-conflictual intergroup context.

Response: We thank Reviewer 1 for their valuable feedback regarding the classification of our research context as non-conflictual. We understand the concern that football rivalries can, at times, escalate into conflicts, and we appreciate the opportunity to clarify our rationale for considering our study context as competitive rather than conflictual.

To address this, we have revised our manuscript to provide a more explicit distinction between competitive and conflictual intergroup settings. In particular, we now highlight how our study differs from prior research on intergroup bias in conflictual contexts (e.g., Jetten et al., 1997; Çoksan & Cingöz-Ulu, 2022), where participants directly allocated resources between two opposing groups (e.g., Kurds vs. Turks). In contrast, in our study, participants allocate resources to their own team, with the remaining funds being distributed among all other teams in the league. Given that their team is not in direct and realistic competition with the majority of these teams, this allocation process does not align with the characteristics of intergroup conflict.

We have also expanded our discussion under the newly added section “Sampling Real-life Group Members in a Competitive Context” to further elaborate on why we do not classify this setting as conflictual. Moreover, we have incorporated an additional limitation acknowledging that certain subsets of football fans experience heightened tensions and that future research may explore how ingroup norms function in highly conflictual fan interactions.

These revisions ensure a clearer articulation of our reasoning and contextual framework. We appreciate the reviewer’s insightful comment, which has helped refine our discussion of the competitive nature of our study context.

Hence, we created the Sampling Real-life Group Members heading and revised the relevant contents on page 6 as follows:

We sampled non-WEIRD real-life group members (football team fans) contrary to many studies in the social identity approach; thus, we aim to extend the findings on the association between ingroup identification and ingroup bias to real and competitive groups in a non-conflictual context. In this way, we thought that we could both increase the generalizability of the findings and contribute to the understanding of cultural dynamics by filling this sampling gap in the literature and thus providing valuable information for practical applications.

Unlike previous studies that examined intergroup bias in conflictual settings [13-14], where resource allocation occurred directly between two competing groups (e.g., Kurds vs. Turks), our study did not position participants in a direct intergroup competitive framework. Instead, participants allocated resources to their own group while the remaining funds were distributed among all teams in the league. Given that their team is not in realistic competition with the majority of these teams, this allocation process lacks the hallmarks of intergroup conflict. While competition among football teams exists, it does not constitute a conflictual intergroup setting in the way that ethnic or political rivalries do. To reiterate, in our study, participants do not allocate resources between their team and a direct rival, but rather between their team and the general league, which consists of many teams with whom their team has no meaningful rivalry. This distinction is crucial in understanding why we do not classify this as a conflictual intergroup context.

 Moreover, in Türkiye, being a football fan is a very strong and consistently prominent form of social identity. Fan identity plays an important social role in shaping social relations and group loyalties not only during matches but also in daily life such as reading about the matches and buying licensed merchandise [29]. For many individuals, loyalty to soccer teams creates a lifelong sense of belonging, and this identity is particularly strong among fans of major teams [30]. Therefore, the identity of football fans in Türkiye provides a relevant and powerful context for the study of group norms and identification processes.

3. Furthermore please note that, on page 13, I would expect an explanation about the differential effects of descriptive and injunctive norms. Otherwise, the claim that injunctive norms might be more relevant to group identity than descriptive norms seems not well justified.

Response: We thank Reviewer 1 for their comment regarding the need for further justification of why injunctive norms might be more relevant to group identity than descriptive norms.

We acknowledge that the original discussion on page 13 required a clearer distinction between these two types of norms (On a related note, please also see our response to point 21 raised by Reviewer 2). We have expanded our explanation by emphasizing that injunctive norms–defined as the moral or socially endorsed ought to rules of behavior–play a fundamental role in defining and reinforcing group behaviors. Unlike descriptive norms, which simply reflect what people do, injunctive norms signal the collective values and moral expectations of a group. Since individuals are motivated to conform to these norms to maintain their social identity and affiliation, injunctive norms exert a stronger influence on group-relevant outcomes. The emphasis on injunctive, rather than descriptive, norms, however, is only important insofar as we want it to influence group-level outcomes.

This contention is supported by prior research demonstrating that when individuals perceive a strong injunctive norm within their ingroup, they experience greater normative pressure to align their behavior accordingly, as failing to do so may threaten their standing within the group (e.g., Hogg & Reid, 2006; Cialdini et al., 1991). We have now made this theoretical connection more explicit in the revised manuscript. Additions made read as follows on page 16:

To investigate this, we employed a novel methodology of manipulating ingroup norms through a group discussion. While our research question did not concern the differential effects of descriptive and injunctive norms, tapping into the latter was necessary for norm manipulation. This is because injunctive norms, representing what ought to be done, exert a significant and consistent influence on emotional and behavioural responses toward rival outgroups, whereas the effects of descriptive norms, representing common behaviours, are less reliable [63]. Therefore, injunctive norms might be more relevant to group identity than descriptive ones [49]. When individuals perceive a strong injunctive norm within their ingroup, they experience greater normative pressure to align their behaviour accordingly, as failing to do so may threaten their standing within the group [39, 64]. This distinction is crucial, as the belief that people should allocate resources equally between ingroup and outgroup members is deemed as an injunctive norm rather than a descriptive one [65]. This means that while people may endorse intergroup equality as the morally right thing to do, they do not necessarily expect it to be widely practised. This approach aligns with the conceptualization of injunctive norms in earlier work [64, 66], where social expectations are communicated through consensus and perceived obligation to the ingroup rather than mere behavioural trends observed within the ingroup.

Finally, we had written an additional paragraph indicating when people would and would not conform to descriptive norms, and another paragraph making the relationship between group identity and injunctive norms more explicit, but we removed those two paragraphs because that version of the article would have been problematic in terms of word count. However, we would like to inform you that we are willing to add them if Reviewer 1 asks us to do so.

4. The authors have summarized literature suggesting that social norms are also formed and negotiated via social interactions in group discussions, emphasizing the active role of each group member. While I find the arguments convincing, and at the same time I have appreciated the novel and creative experimental manipulation of social norms, built on classic research, I think there is a mismatch between the theoretical rationale and the actual experimental manipulation. Indeed, there were no real group discussions, and therefore there was no negotiation. Each participants was merely invited to express their opinion after being exposed to the opinion of three other alleged ingroup members. I would encourage the authors to carefully think about the best justification for their method, and to avoid overemphasizing some strengths which might not fully apply to their method.

Response: We thank Reviewer 1 for their thoughtful feedback and for recognizing both the strength of our theoretical rationale and the novel nature of our experimental manipulation. We appreciate your reservations around the alignment between our theoretical framework and experimental manipulation. By rights, we acknowledge that our manipulation did not involve a group discussion in which participants actively negotiated norms through back-and-forth interactions. However, we maintain that our approach still captures a key mechanism of social norm formation–norm transmission via group consensus and social exposure–which aligns with established research on how norms emerge and exert influence within groups (Cialdini et al., 1991; Hogg & Reid, 2006; Moscovici, 1974).

Unfortunately, standardizing the manipulation required

---

## [Decision Letter · Decision Letter 1]

27 May 2025

PONE-D-24-50337R1
Egalitarian Norms Can Deflate Identity-Bias Link in Real-life Groups
PLOS ONE

Dear Dr. Çoksan,

Thank you for submitting your manuscript to PLOS ONE. After careful consideration, we feel that it has merit but does not fully meet PLOS ONE’s publication criteria as it currently stands. Therefore, we invite you to submit a revised version of the manuscript that addresses the points raised during the review process.
 
Both reviewers have appreciated your efforts in revising the manuscript. However, both of them raised problems that they found not addressed in the revised manuscript. To avoid redundancy, please see their comments and address their points one by one.

We look forward to receiving your revised manuscript.

Kind regards,

I-Ching Lee

Academic Editor

PLOS ONE

Reviewers' comments:

Reviewer's Responses to Questions

**Comments to the Author**

1. If the authors have adequately addressed your comments raised in a previous round of review and you feel that this manuscript is now acceptable for publication, you may indicate that here to bypass the “Comments to the Author” section, enter your conflict of interest statement in the “Confidential to Editor” section, and submit your "Accept" recommendation.

Reviewer #1: (No Response)

Reviewer #2: (No Response)

2. Is the manuscript technically sound, and do the data support the conclusions?

Reviewer #1: Partly

Reviewer #2: Yes

3. Has the statistical analysis been performed appropriately and rigorously? 

Reviewer #1: (No Response)

Reviewer #2: Yes

4. Have the authors made all data underlying the findings in their manuscript fully available?

Reviewer #1: Yes

Reviewer #2: Yes

5. Is the manuscript presented in an intelligible fashion and written in standard English?

Reviewer #1: (No Response)

Reviewer #2: Yes

6. Review Comments to the Author

Reviewer #1: The authors have made considerable efforts to improve the manuscript by taking into account all the suggestions provided by the reviewers. Furthermore, as stated in my previous review, I think that this research deserves to be published – possibly in a high impact journal such as Plos One.

Besides these premises, unfortunately the current version of the article is still not fully clear, some sections and arguments are not fully convincing, and, most importantly, there is still ambiguity in the reported results.

1. Major issue:

In Study 2, the authors have deleted 2 X 2 ANOVA results, and rather just kept findings with measured identification and norms. I am aware that I made a comment about inconsistency in reporting findings which has probably been misinterpreted and which has led to this choice, and I am sorry about it. However, in my view this choice makes no sense. Study 2 is an experiment, and main data analyses need to focus on effects of experimentally manipulated variables, i.e. identification and norms. Otherwise, why conducting an experiment? I would strongly recommend that the authors report findings of the effects of experimentally manipulated norms, identification, and their interaction. Please note that the discussion includes several sentences about the effects of experimentally manipulated variables (likely kept from the previous version of the discussion) which make no sense as such findings are not reported.

Additional issues:

2. The argument that the experimental manipulation of social norms in Study 2 taps injunctive norms more than descriptive norms is not convincing. Participants read the opinion of other ingroup members (football fans supporting the same team) about how advertising revenues should be distributed, but I guess that in the real world such distribution is determined by the league and by negotiations between the league and the football clubs, with very limited voice by football fans. Therefore, the norms experimental manipulation is not much about how ingroup members (football fans supporting the same team) should act but rather about what ingroup members think. In my view this aligns more with descriptive rather than injunctive norms.

Relatedly, the authors are still referring to the experimental manipulation as “interactive” repeatedly throughout the manuscript. There was no actual interaction, because participants just read the opinion of others and expressed their opinion. While I find the experimental manipulation through a simulated discussion very thoughtful and creative, it should not be described as “interactive”.

3. The argument that the intergroup context is not conflictual is still not fully convincing. I agree that the distribution of resources task used in the outcome variable is not conflictual, but relationships between fans of different football teams can be conflictual.

4. Regarding the organization of the introduction, I think that the authors have improved it a lot compared to the previous version. However, readability could be further improved. I would recommend that the authors clearly define the following concepts, before mentioning them for the first time: differentiation, favoritism, fairness, egalitarianism. While I agree in the distinction between fairness and egalitarianism, and that egalitarianism might be a stronger means to counteract intergroup bias than fairness, the rhetoric on the distinction between the two could be clearer and more straightforward to follow.

5. The general discussion needs to be thoroughly revised after the implementation of the other changes, especially point 1.

Minor issues:

6. Study 2 sample size might not be ideal given the 2 X 2 research design. It is also still unclear which power analysis was conducted.

7. When reporting decomposition of interactions, have the authors calculated simple effects at +1 and -1 standard deviation of the moderator? This needs to be clarified in the main text.

8. I do not agree with the interpretation of the mid-point of the social norms measure. Unless there was a label associated to the mid-point, it cannot be surely known that “the midpoint does not represent a neutral position but rather an intermediate perception that ingroup members sometimes act in accordance with egalitarianism and sometimes with favouritism”. It is also possible that the mid-point is interpreted as a neutral opinion. Please note that all the lines on the mid-point might not be necessary (see next comment).

9. While Plos One might have no word limit, I think that some sections could be shortened to make the article easier to read and follow. For example, in the introduction there is one page and a half on what happens to people with low identification, but such text sounds partly repetitive. Please note that the comment on the length of the paper is determined also by the fact that this research largely replicates the theoretical rationale, hypotheses and findings by Jetten et al. (1997) and by Çoksan and Cingöz-Ulu (2022). I believe that this research program deserves to be published, the article could be shorter and, most importantly, more straightforward.

10. Finally, there might still be some inaccuracies. For example, the sentence “the relationship between ingroup identification and ingroup bias remains present even at lower levels of identification” is imprecise. The whole article should be carefully re-checked after resubmission.

Reviewer #2: I would like to thank the authors for a much improved manuscript.

I have only a few comments.

- While the authors corrected the usage of the word "neutral" when referring to mid-value of their ingroup favoritism measure, the word stays in the figures. Please change the wording in the figures as well for a correct description.

- Please include in the manuscript, maybe in the Appendix, a few example discussions generated in Study 2. Any reader is left blind as to its contents.

- Describe in text what high and low refer to in the figures; are these the extremes, or grouped values? If the latter, what are the cutoffs?

- "completely egalitarianism" should be "complete egalitarianism", and likewise, completely favouritism should be complete favouritism.

- In the first paragraph of the introduction, "In such resource allocation tasks, the perception of resource scarcity appears to promote the endorsement of zero-sum beliefs and, in turn, favouring the ingroup over outgroups [4].", I do not see how this is relevant to this work, especially in the opening paragraph. The resource allocation tasks are zero-sum, so referring to this finding generates a misleading idea of what the research is about.

- Change should to would or are expected to be in the following sentence, as you are making a prediction not a prescription for behavior: "These findings indicate that people with high ingroup identification should be more mindful of ingroup norms and accordingly revise their behaviour than people with low ingroup identification."

- "one of the few studies", not "one of the rare studies".

- Sentences with "we thought that" read unnatural and not particularly scientific. These words can be omitted.

- I do not see how the study introduces cultural dynamics, and the following claim can be made: "In this way, we thought that we could both increase the generalizability of the findings and contribute to the understanding of cultural dynamics by filling this sampling gap in the literature."

- I am not sure having different values mean in the following sentence, please specify or change wording: "Participants randomly assigned to a low ingroup identification condition, on the other hand, were informed that they have different values from their team’s other fans and were an atypical member of their ingroup."

- Please complete this phrase after "to a lesser extent" indicating the comparison unit "Our main expectation that those with higher identification in the egalitarian norm would favour their ingroup to a lesser extent was not supported in either of the studies (H2)"

- How can the authors be sure that they did not draw ultra-fans to their study? The discussion makes such a claim.

7. PLOS authors have the option to publish the peer review history of their article (what does this mean?). If published, this will include your full peer review and any attached files.

Reviewer #1: No

Reviewer #2: No

---

## [Author Response · Author response to Decision Letter 2]

11 Jun 2025

Response Letter

Dear Editor,

We are grateful to reviewers’ constructive feedback on our paper. To address reviewers’ concerns, first, we reorganized and shorten introduction and general discussion. Second, we have added the ANOVA results to Study 2 and re-checked the General Discussion heading to ensure that it aligns with all of these findings. Below, we have re-numbered each comments of Reviewers and provided our answers. We hope that this version of the paper is suitable for the journal.

Reviewer #1

The authors have made considerable efforts to improve the manuscript by taking into account all the suggestions provided by the reviewers. Furthermore, as stated in my previous review, I think that this research deserves to be published – possibly in a high impact journal such as Plos One. Besides these premises, unfortunately the current version of the article is still not fully clear, some sections and arguments are not fully convincing, and, most importantly, there is still ambiguity in the reported results.

1. In Study 2, the authors have deleted 2 X 2 ANOVA results, and rather just kept findings with measured identification and norms. I am aware that I made a comment about inconsistency in reporting findings which has probably been misinterpreted and which has led to this choice, and I am sorry about it. However, in my view this choice makes no sense. Study 2 is an experiment, and main data analyses need to focus on effects of experimentally manipulated variables, i.e. identification and norms. Otherwise, why conducting an experiment? I would strongly recommend that the authors report findings of the effects of experimentally manipulated norms, identification, and their interaction. Please note that the discussion includes several sentences about the effects of experimentally manipulated variables (likely kept from the previous version of the discussion) which make no sense as such findings are not reported.

Response: We sincerely thank Reviewer 1 for this important and constructive comment. Upon reflection, we realize that we had misunderstood their previous remark regarding inconsistencies in reporting findings, which led us to remove the 2 x 2 ANOVA from Study 2 in the revised version. We apologize for this misinterpretation and for the unintended consequence of weakening the alignment between our experimental design and the statistical analyses reported. As they rightly noted, Study 2 employed an experimental design in which both ingroup identification and ingroup norm were manipulated. We completely agree that the primary analyses in such a design should focus on the effects of these experimentally manipulated variables. In response to your current feedback, we have now re-included the full 2 x 2 between-subjects ANOVA in the Results section of Study 2.

This analysis tested the main and interaction effects of manipulated ingroup identification (high vs. low) and ingroup norm (favouritism vs. egalitarianism) on ingroup bias. The results showed significant main effects for both manipulated ingroup identification and manipulated ingroup norm, indicating that participants with high ingroup identification exhibited more ingroup bias than those in the low identification condition, and participants under the favouritism norm condition showed more ingroup bias than those under the egalitarianism norm condition.

Although the interaction effect was not statistically significant, we conducted pre-planned post-hoc comparisons to further explore the potential moderating role of identification, as hypothesized. At this point, we would like to emphasize one issue. We are familiar with the fact that post-hoc comparisons are generally not performed if an interaction effect is not statistically significant, but we do not disregard the recommendations in the literature that post-hoc comparisons should be conducted even if they are not statistically significant, especially if there is an expected interaction or if it is considered theoretically important. In our study, the statement that post-hoc comparisons were performed to identify a possible moderating role clearly indicates that there is a theoretical justification for this approach. In other words, as expected in the literature (see Çoksan & Cingöz Ulu, 2022; Jetten et al., 1997), we performed post-hoc comparisons for the interaction, even if it was not statistically significant, in order to further examine a possible moderating role. We would like to emphasize this justification again here. In conclusion, these comparisons revealed that in both high and low identification conditions, participants exposed to the favouritism norm showed significantly more ingroup bias than those exposed to the egalitarian norm. Moreover, under the favouritism norm condition, participants with high ingroup identification displayed more ingroup bias than those with low identification, whereas no significant difference was observed under the egalitarian norm condition.

In light of these findings, we have also thoroughly revised the General Discussion heading. Specifically, we removed or rephrased any statements that referenced experimental effects without supporting results in the previous version. The current version now accurately reflects the re-inserted statistical findings and draws conclusions that are fully grounded in the reported analyses. We also wish to highlight that the decision to conduct these post-hoc comparisons was not data-driven but theoretically justified and planned in advance, as our hypotheses involved the moderation of identification by group norms. We have clarified this rationale in the Methods and Results sections as well.

We are grateful for Reviewer 1’s feedback, which significantly improved the methodological integrity and reporting transparency of our manuscript. Their input helped us ensure that our statistical reporting is appropriately aligned with our experimental design and theoretical aims. If any additional clarifications or improvements are needed, we would be happy to address them as well.

2a. The argument that the experimental manipulation of social norms in Study 2 taps injunctive norms more than descriptive norms is not convincing. Participants read the opinion of other ingroup members (football fans supporting the same team) about how advertising revenues should be distributed, but I guess that in the real world such distribution is determined by the league and by negotiations between the league and the football clubs, with very limited voice by football fans. Therefore, the norms experimental manipulation is not much about how ingroup members (football fans supporting the same team) should act but rather about what ingroup members think. In my view this aligns more with descriptive rather than injunctive norms.

Response: We appreciate the reviewer’s attention to the distinction between injunctive and descriptive norms, and we think that they often overlap in real-life settings. We do not claim that our manipulation taps purely or exclusively into injunctive norms. In fact, we acknowledge that elements of both descriptive and injunctive norm communication are embedded in our design; as is often the case when group members express their views about a behavior’s typicality and the necessity to endorse it. For instance, we asked participants whether they believed that “most of the people I spoke to think that we should treat all parties equally in the allocation of resources such as advertising revenues,” capturing both perceived consensus (a descriptive element) and prescriptive belief (an injunctive element).

That said, we maintain that our manipulation aligns more closely with injunctive norms for 4 reasons:

a. In both egalitarianism and favouritism conditions, the chat structure explicitly frames the conversation around what ingroup members should do. The moderator’s questions repeatedly ask for judgments of what the Turkish Football Federation ought to do with revenue allocation, and more importantly, whether participants believe that supporters of their team value that approach. This framing encourages reflection on collective moral obligation, not just observed behavior.

In participants’ responses, this moralizing obligation on the group is evident. In the egalitarian condition, for example, participants say that “giving more share to a team does not befit our fans” and “equal shares means equal fighting power.” Similarly, in the favouritism condition, participants state that “you’re hurting your own team” if you don't advocate for more funding to your team, and that “In terms of fair play, it is important that such a resource is transferred more to the teams that create that resource.” These are not mere reports of common practice but expressions of what is appropriate or expected from group members; inducing a sense that it is the right thing to do for the ingroup.

b. We respectfully disagree with the characterization of the manipulation as participants “reading the opinion of other ingroup members.” This interpretation overlooks critical aspects of the procedure that distinguish it from a passive exposure paradigm (e.g., reading a forum or article). In our design, participants took part in a time-paced, simulated group chat moderated by a researcher. They were called to engage with the conversation three times by contributing their own views after seeing how other ingroup members had already articulated a stance. This setting mimics an interactive group discussion, emphasizing active participation rather than observational learning.

The chat structure itself promotes consensus-building. Participants responded in a fixed order, allowing the first speaker to establish a normative tone that subsequent speakers then addressed and affirmed and elaborated upon, instead of speaking up their mind with no attention to the previous speaker. In the egalitarian condition, for instance, the second speaker not only showed agreement with the first speaker's response but provided additional justifications rooted in group values (e.g., fairness, equality, collective responsibility). A parallel structure was observed in the favouritism condition, where speakers reinforced the moral importance of supporting their own team. This structure reflects how injunctive norms are formed and maintained: through group discussions that converge on shared expectations via moralizing group reasoning.

c. We agree with the reviewer that football fans have little to no formal control over the actual distribution of money between football clubs. However, this lack of direct influence does not undermine the function of injunctive norms. Actions, even those that appear irrational from a purely instrumental perspective (e.g., minimal group paradigm), can serve to affirm group membership and group’s positive distinctiveness.

Normative influence, especially injunctive norms, does not depend on individuals’ ability to directly affect policy or outcomes. Even when people cannot materially change a situation, they often express what they believe their group should do (Jaeger & Schultz, 2017). In our study, participants routinely refer to group-based ideals (e.g., “this is what our fans value,” “it’s important for fair play”), indicating an orientation toward group-based moral consensus rather than strategic efficacy. The normative expectations they voiced were less about influencing real-world policy and more about signaling identity-based commitments and obligations within their ingroup. In fact, as previous research has shown, injunctive group norms can emerge precisely in contexts where individuals rely on group-based moral discourse rather than perceived efficacy to influence outcomes (e.g., Acar et al., 2024; Workman et al., 2020). This indicates that normative influence can stem from internalized group norms rather than the ability to directly influence policy.

Thus, while fans may lack direct power over league decisions, their alignment with injunctive norms remains meaningful and, we argue, central to how the manipulation operates.

d. The content of the conversations in our experiment revolves around fairness, deservingness, and group values, all of which are central to injunctive norm perception (Heuver et al., 1999; Juan-Bartroli, 2024). Crucially, this emphasis holds regardless of the direction of the normative stance. Whether participants advocate for equal distribution or favoring their own team, their justifications are rooted in what they believe to be right, deserved, or morally appropriate, more than in what is merely common or typical.

In sum, while the manipulation incorporates both norm elements, we believe it is more aligned with injunctive norms due to its structure and content. Considering the request to further shorten the manuscript, we have opted not to include these additional explanations in the main text.

In conclusion, while the chat content reflects the perceived views of ingroup members, the structure of the questions and participants’ own responses go beyond mere observation of others’ opinions. The wording used and the moral reasoning in their responses indicate not just perceived typical behavior, but shared group obligations. Therefore, the manipulation invites participants to deliberate on what is appropriate and expected, not merely on what is common. For instance, the contents that are used during the ingroup norm manipulation task were as follows:

MODERATOR DENIZ:

Hello, I’m Deniz, the researcher, and I’ll be moderating this discussion. My request is that you answer only the questions I ask, in the specified order. For each question, please respond in this sequence: 1) Kraker1, 2) sonütücü, 3) Kalemşör, 4) ${q://QID91/ChoiceTextEntryValue}. You may need to scroll down to see what I’m about to write, as it may appear at the bottom of your browser.

MODERATOR DENIZ:

I’ll now ask you some questions about how you think the Turkish Football Federation (TFF) should distribute its advertising revenue. But first, I’d like to provide some background information on TFF’s revenue streams.

Once everyone is ready, you can click the "Next" button that will appear at the bottom right.

MODERATOR DENIZ:

The Turkish Football Federation is the sole authorized organization to sell broadcasting rights for matches of teams competing in the Turkish Süper Lig to TV channels, radio stations, and social media platforms. The federation generates income by selling these broadcasting rights. Additionally, they receive a share of the revenue from product advertisements shown to consumers during matches. With this substantial income, TFF distributes a portion of the revenue among the league’s teams.

MODERATOR DENIZ:

This distribution process sometimes sparks debate. Some fans argue that all teams should receive equal shares, while others believe more investment should go to financially disadvantaged teams to foster football development. On the other hand, some fans claim that teams with higher viewership should receive larger shares since their matches attract more attention. This revenue is one of the income sources for the league’s teams.

MODERATOR DENIZ:

My first question is: How do you think TFF should distribute this advertising revenue? You may answer in order.

Kraker1:

I believe our team’s fans value equality in this matter. Whether it’s a big club or an Anatolian team, it shouldn’t matter. This revenue comes from all teams, so it should be distributed equally.

sonütücü:

I think similarly. Yes, some teams get more coverage in newspapers and on TV, but private companies already direct their ads to those teams. Other revenue exists because there’s a league. If the league is a whole and the teams are its stakeholders, at least this income should be shared equally.

Kalemşör:

I also support equal distribution. Why should one team get a larger share? Even if some teams appear more in ads, those ads aren’t only watched by their fans—the revenue comes from all viewers. Besides, favoring one team, whether ours or not, wouldn’t sit right with our fanbase.

${q://QID91/ChoiceTextEntryValue} IT’S YOUR TURN. AFTER WRITING YOUR ANSWER IN THE BOX BELOW, YOU CAN CLICK THE "NEXT" BUTTON THAT WILL APPEAR AT THE BOTTOM RIGHT.

MODERATOR DENIZ:

Thank you for your responses. Do you think equal

---

## [Decision Letter · Decision Letter 2]

22 Jul 2025

PONE-D-24-50337R2
Egalitarian Norms Can Deflate Identity-Bias Link in Real-life Groups
PLOS ONE

Dear Dr. Çoksan,

Thank you for submitting your manuscript to PLOS ONE. After careful consideration, we feel that it has merit but does not fully meet PLOS ONE’s publication criteria as it currently stands. Therefore, we invite you to submit a revised version of the manuscript that addresses the points raised during the review process.

We look forward to receiving your revised manuscript.

Kind regards,

I-Ching Lee

Academic Editor

PLOS ONE

Journal Requirements:

Reviewers' comments:

Reviewer's Responses to Questions

**Comments to the Author**

1. If the authors have adequately addressed your comments raised in a previous round of review and you feel that this manuscript is now acceptable for publication, you may indicate that here to bypass the “Comments to the Author” section, enter your conflict of interest statement in the “Confidential to Editor” section, and submit your "Accept" recommendation.

Reviewer #1: (No Response)

2. Is the manuscript technically sound, and do the data support the conclusions?

Reviewer #1: Yes

3. Has the statistical analysis been performed appropriately and rigorously? 

Reviewer #1: Yes

4. Have the authors made all data underlying the findings in their manuscript fully available?

Reviewer #1: Yes

5. Is the manuscript presented in an intelligible fashion and written in standard English?

Reviewer #1: Yes

6. Review Comments to the Author

Reviewer #1: The authors have thoroughly revised the article by taking into account the second round of comments by the reviewers. While the article is considerably improved, there are still some minor issues which need to be addressed.

1. The definition of fairness on page 3 is unclear and does not fully match what is stated later in the article about fairness.

2. Study 2 findings. The wording when reporting the decomposition is not fully clear. For example, the sentence “Post-hoc comparisons between experimental conditions indicated that participants with favoritism ingroup norm show more ingroup bias (N = 30, M = 5.21, SE = .93) than those with egalitarianism norm condition (N = 31, M = 2.15, SE = .92; p = .021, 95% CI [.476, 5.645]) on low ingroup identification condition.” Could be revised as something like “Post-hoc comparisons indicated that, among participants in the low ingroup identification condition, there was more ingroup bias among those in the favoritisms ingroup norm condition compared to those in the egalitarianism ingroup norm condition.” The following sentences should be revised as well for clarity and correctness.

3. Still on Study 2 findings. As previously explained, the main moderation hypothesis needs to be tested with ANOVA and post-hoc comparisons. However, the authors have chosen to keep results reporting the interaction tested with cross-sectional data. If the authors want to keep that section, they need to clarify the goal of such analysis, and specifically how they justify a moderation test run with measures which were initially intended to act as manipulation checks.

4. Study 2 Discussion. English language still needs to be revised. The second sentence begins with “But unlike Study 1, Study 2 revealed in this study we observed…” which is not correct. Overall, once more, the whole article should be more carefully checked.

5. Both in the Study 2 discussion and in the general discussion, the authors cannot state that their effect of experimentally manipulated ingroup norms is STRONG (or stronger), as they have not conducted a test on the strength of the effect(s). They should merely state that the effect is significant.

6. While I get the arguments by the authors, I still feel that the (long) text distinguishing between descriptive and injunctive norms has limited relevance, given that the experimental manipulation does not focus exclusively and univocally on injunctive norms. That said, I have repeated more or less the same comment three times, and it is up to the editor to decide whether this change is necessary or not.

7. PLOS authors have the option to publish the peer review history of their article (what does this mean?). If published, this will include your full peer review and any attached files.

Reviewer #1: No

---

## [Author Response · Author response to Decision Letter 3]

30 Jul 2025

Response Letter

Dear Editor,

We thank the Reviewer for their feedback. Below, we have numbered and responded to each of the six comments. We believe we have addressed all of the Reviewer’s concerns. To do so, we first provided a more detailed definition of the concept of fairness in the section where it is first introduced, ensuring consistency with the rest of the text. We revisited the points raised by the Reviewer regarding the reporting of results. In Study 2, we clearly stated our rationale for conducting a moderation analysis alongside the ANOVA to test our hypotheses. Finally, we carefully reviewed the manuscript sentence by sentence to ensure the language is flawless. We hope that the revised version is now suitable for publication.

Reviewer #1

The authors have thoroughly revised the article by taking into account the second round of comments by the reviewers. While the article is considerably improved, there are still some minor issues which need to be addressed.

1. The definition of fairness on page 3 is unclear and does not fully match what is stated later in the article about fairness.

Response: We thank Reviewer 1 for highlighting this issue. We compared the initial mention of the fairness norm on Page 3 with the section titled “Fairness vs. Egalitarianism” to ensure consistency in how the norm is defined. To make the first usage consistent with the more detailed explanation later in the manuscript, we briefly and clearly introduced the core elements included in that later definition, such as merit, proportionality, and need. This adjustment also helps clarify the flexible and context-dependent nature of the fairness norm. We believe that this revision ensures consistency throughout the manuscript. The revised statement on Page 3 now reads as follows:

The difference between people with high and low identification, however, disappeared when the fairness norm, which refers to collective standards within a group regarding what its members consider right, appropriate, or justified based on criteria such as merit, proportionality, or need, was in place. Fairness norms are thus context-dependent and may lead to unequal outcomes that are nonetheless perceived as legitimate or justified by group members.

2. Study 2 findings. The wording when reporting the decomposition is not fully clear. For example, the sentence “Post-hoc comparisons between experimental conditions indicated that participants with favoritism ingroup norm show more ingroup bias (N = 30, M = 5.21, SE = .93) than those with egalitarianism norm condition (N = 31, M = 2.15, SE = .92; p = .021, 95% CI [.476, 5.645]) on low ingroup identification condition.” Could be revised as something like “Post-hoc comparisons indicated that, among participants in the low ingroup identification condition, there was more ingroup bias among those in the favoritisms ingroup norm condition compared to those in the egalitarianism ingroup norm condition.” The following sentences should be revised as well for clarity and correctness.

Response: We thank Reviewer 1 for pointing out the need for greater clarity. We have revised the sentences in the Study 2 findings section as suggested. In this revised version, we simplified the sentence structure and clarified which groups were compared under which conditions. We emphasized the experimental ingroup norm conditions (favoritism norm, egalitarianism norm) and levels of identification (high identification, low identification). This allowed us to present the results more clearly and made the sentences easier to follow. These adjustments significantly enhance readability and ensure that our findings are accurately conveyed. The final version of the four sentences highlighted by Reviewer 1 appears on page 21 as follows:

Post-hoc comparisons indicated that, among participants in the low ingroup identification condition, there was more ingroup bias in the favoritism norm condition (N = 30, M = 5.21, SE = .93) compared to the egalitarianism norm condition (N = 31, M = 2.15, SE = .92; p = .021, 95% CI [.476, 5.645]). Similarly, among participants in the high ingroup identification condition, those in the favoritism norm condition (N = 32, M = 8.33, SE = .90) displayed greater ingroup bias compared to participants in the egalitarianism norm condition (N = 41, M = 2.86, SE = .80; p < .001, 95% CI [3.092, 7.852]). In the favoritism norm condition, participants with high ingroup identification (M = 8.33, SE = .90) showed more ingroup bias compared to those with low ingroup identification (M = 5.21, SE = .93; p = .017, 95% CI [.558, 5.686]). However, in the egalitarianism norm condition, there was no significant difference in ingroup bias between participants with high and low ingroup identification (p = .559, 95% CI [-1.690, 3.113]).

3. Still on Study 2 findings. As previously explained, the main moderation hypothesis needs to be tested with ANOVA and post-hoc comparisons. However, the authors have chosen to keep results reporting the interaction tested with cross-sectional data. If the authors want to keep that section, they need to clarify the goal of such analysis, and specifically how they justify a moderation test run with measures which were initially intended to act as manipulation checks.

Response: We thank Reviewer 1 for highlighting this important methodological issue. To clarify, the moderation analysis conducted with continuous scores on perceived ingroup identification and perceived ingroup norms (initially used as manipulation checks) was included to provide convergent validity and methodological triangulation of our experimental findings rather than independently testing our main hypotheses. Specifically, we now explicitly emphasize in the revised manuscript that this supplementary analysis:

(a) Serves primarily as an additional verification method to explore the robustness of our experimental results through an alternative analytic approach (see Hayes, 2022); (b) Allows us to examine how individual differences in participants’ subjective perceptions in response to manipulations relate to ingroup bias, thus enhancing our understanding of the moderation process; and (c) Provides methodological consistency and comparability between Study 1 (correlational) and Study 2 (experimental), enabling us to offer a more integrative narrative across both studies.

This clarification has been explicitly included in the Results section of Study 2, immediately following the ANOVA results and before presenting the supplementary moderation analysis. We greatly appreciate Reviewer 1’s insightful comment, as addressing it allowed us to enhance the methodological transparency and analytic coherence of our manuscript. The paragraph we added on Page 22 is as follows:

To provide convergent validity and methodological triangulation of our experimental findings, we additionally conducted a supplementary moderation analysis using participants’ continuous scores on the manipulation checks of perceived ingroup identification and perceived ingroup norms. It is important to clarify that although these measures initially served as manipulation checks, employing them in a moderation model allows us to examine the robustness of our experimental results through an alternative analytic strategy [65]. Specifically, while the ANOVA approach tested the causal impact of manipulated categorical variables, this complementary moderation analysis explores how individual differences in response to manipulations may be systematically related to ingroup bias. This supplementary analysis was preplanned and serves to support the primary experimental analyses rather than to independently test our hypotheses. Moreover, the moderation model conducted in Study 2 parallels the analytical approach employed in Study 1. Thus, presenting this analysis allows for direct methodological consistency and comparability across studies. Specifically, it enables us to assess whether the moderation observed through measured continuous variables in Study 1 replicates conceptually under experimental manipulation conditions in Study 2, thereby offering a more integrative and robust narrative across both studies.

4. Study 2 Discussion. English language still needs to be revised. The second sentence begins with “But unlike Study 1, Study 2 revealed in this study we observed…” which is not correct. Overall, once more, the whole article should be more carefully checked.

Response: We reviewed the entire manuscript sentence by sentence to ensure the accuracy of the language. In the track changes version of the manuscript, various edits can be seen. We believe that these revisions have made the manuscript linguistically sufficient.

5. Both in the Study 2 discussion and in the general discussion, the authors cannot state that their effect of experimentally manipulated ingroup norms is STRONG (or stronger), as they have not conducted a test on the strength of the effect(s). They should merely state that the effect is significant.

Response: We identified the sentences containing these expressions one by one and revised them in line with the Reviewer’s suggestions.

6. While I get the arguments by the authors, I still feel that the (long) text distinguishing between descriptive and injunctive norms has limited relevance, given that the experimental manipulation does not focus exclusively and univocally on injunctive norms. That said, I have repeated more or less the same comment three times, and it is up to the editor to decide whether this change is necessary or not.

Response: In this version, we believe that we briefly stated the differentiation between the two norms at the beginning of Study 2, but we are also eager to follow the Editor’s recommendation on this matter and will shorten this part of the manuscript if necessary.

---

## [Editor Report · Decision Letter 3]

4 Aug 2025

Egalitarian Norms Can Deflate Identity-Bias Link in Real-life Groups

PONE-D-24-50337R3

Dear Dr. Çoksan,

We’re pleased to inform you that your manuscript has been judged scientifically suitable for publication and will be formally accepted for publication once it meets all outstanding technical requirements.

Kind regards,

I-Ching Lee

Academic Editor

PLOS ONE
---

## [Editor Report · Acceptance letter]

PONE-D-24-50337R3

PLOS ONE

Dear Dr. Çoksan,

I'm pleased to inform you that your manuscript has been deemed suitable for publication in PLOS ONE. Congratulations! Your manuscript is now being handed over to our production team.

Kind regards,

on behalf of

Dr. I-Ching Lee

Academic Editor

PLOS ONE